# Decoupled Graph Energy-based Model for Node Out-of-Distribution Detection on Heterophilic Graphs

**Yuhan Chen**[1*] **& Yihong Luo**[2*], **Yifan Song**[3], **Pengwen Dai**[4], **Jing Tang**[3,2†], **Xiaochun Cao**[4†]

[1] The School of Computer Science and Engineering, Sun Yat-sen University
[2] The Hong Kong University of Science and Technology
[3] The Hong Kong University of Science and Technology (Guangzhou)
[4] The School of Cyber Science and Technology, Shenzhen Campus of Sun Yat-sen University

## Abstract

Despite extensive research efforts focused on Out-of-Distribution (OOD) detection on images, OOD detection on nodes in graph learning remains underexplored. The dependence among graph nodes hinders the trivial adaptation of existing approaches on images that assume inputs to be i.i.d. sampled, since many unique features and challenges specific to graphs are not considered, such as the heterophily issue. Recently, GNNSafe, which considers node dependence, adapted energy-based detection to the graph domain with state-of-the-art performance, however, it has two serious issues: 1) it derives node energy from classification logits without specifically tailored training for modeling data distribution, making it less effective at recognizing OOD data; 2) it highly relies on energy propagation, which is based on homophily assumption and will cause significant performance degradation on heterophilic graphs, where the node tends to have dissimilar distribution with its neighbors. To address the above issues, we suggest training Energy-based Models (EBMs) by Maximum Likelihood Estimation (MLE) to enhance data distribution modeling and removing energy propagation to overcome the heterophily issues. However, training EBMs via MLE requires performing Markov Chain Monte Carlo (MCMC) sampling on both node feature and node neighbors, which is challenging due to the node interdependence and discrete graph topology. To tackle the sampling challenge, we introduce Decoupled Graph Energy-based Model (DeGEM), which decomposes the learning process into two parts—a graph encoder that leverages topology information for node representations and an energy head that operates in latent space. Additionally, we propose a Multi-Hop Graph encoder (MH) and Energy Readout (ERo) to enhance node representation learning, Conditional Energy (CE) for improved EBM training, and Recurrent Update for the graph encoder and energy head to promote each other. This approach avoids sampling adjacency matrices and removes the need for energy propagation to extract graph topology information. Extensive experiments validate that DeGEM, without OOD exposure during training, surpasses previous state-of-the-art methods, achieving an average AUROC improvement of 6.71% on *homophilic* graphs and 20.29% on *heterophilic* graphs, and even outperform methods trained with OOD exposure. Our code is available at: `https://github.com/draym28/DeGEM`.

## 1 Introduction

Detecting Out-of-Distribution (OOD) data is crucial for enhancing AI models' robustness, reliability, and safety in real-world scenarios, where input data may deviate from the training data distribution. Many works (Hendrycks & Gimpel, 2016; Liang et al., 2018; Lee et al., 2018; Hendrycks et al., 2018; Liu et al., 2020) have been proposed for OOD detection tasks on i.i.d. data, e.g., images. Such an

---

[*]Co-First Authors: Yuhan Chen (draym@qq.com) and Yihong Luo (yluocg@connect.ust.hk).
[†]Corresponding Authors: Jing Tang (jingtang@ust.hk) and Xiaochun Cao (caoxiaochun@mail.sysu.edu.cn).

urgent need also exists in the domains that apply graph-format data, such as medical-diagnosis (Kukar, 2003) and autonomous driving (Dai & Van Gool, 2018). However, there has been a notable lack of work focusing on graph OOD node detection, i.e., detecting OOD nodes on which the model is expected to have low confidence (Amodei et al., 2016; Liang et al., 2018). Due to the interdependence among graph nodes, it is hard to apply the methods designed for i.i.d. inputs directly.

Recently, GNNSafe (Wu et al., 2023a) adapted energy-based OOD detection for graph data mainly by using GNNs as the backbone, and output energy for nodes as confidence, achieving current state-of-the-art performance. However, it is trained by classification loss and constructs node energy by classification logits without specifically designed training for modeling data distribution, limiting the performance of identifying OOD nodes. Moreover, the energy propagation technique, used in GNNSafe, heavily relies on the homophily assumption, i.e., neighbors often belong to the same class. This will lead to a significant performance degradation on heterophilic graphs, where neighbors do not share a similar distribution. To mitigate these issues, we suggest training Energy-based Models (EBMs)

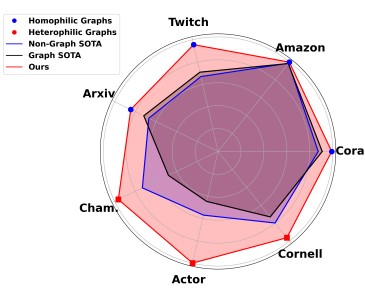

Figure 1: AUROC across graphs.

to detect OOD instances via Maximum Likelihood Estimation (MLE) to obtain better energy, and removing energy propagation to address the heterophily issue. However, training EBMs via MLE requires performing Markov Chain Monte Carlo (MCMC) sampling in training, which is notorious for graphs due to the complexity of graph topology.

In this paper, our proposed **De**coupled **G**raph **E**nergy-based **M**odel (DeGEM) alleviates both the heterophily issue and sampling challenges of learning EBMs for large graphs. The *key insight* in our work is: GNNs can extract the topology information, forming latent space without interdependence. Hence, we can conduct MCMC sampling on latent space to train the energy head. This approach is computationally efficient, as it avoids sampling the adjacency matrix, and eliminates the need for energy propagation techniques, which could degrade performance on heterophilic graphs. Specifically, DeGEM decomposes the EBM into two components: graph encoder and energy head. First, the graph encoder is trained by the Graph Contrastive Learning (GCL) algorithm and classification loss for obtaining informative node representations, then the energy head, trained over latent space by MLE, outputs the energy scores of nodes, which are used as nodes' OOD scores. By doing so, DeGEM transposes operations inherently dependent on the graph structure into the latent representation domain, thereby decoupling subsequent steps from the graph structure's dependency. The benefits are twofold: 1) MCMC sampling can be efficiently conducted on low-dimensional latent space to sample node representations only, dramatically decreasing the computation cost; 2) the Energy head does not require a propagation operation to further extract topology information, avoiding performance degradation on heterophilic graphs.

Moreover, to better unleash the effectiveness of DeGEM in node OOD detection, we propose several principled training designs: a Multi-Hop Graph encoder (MH) and Energy Readout (ERo) to enhance node representation learning, Conditional Energy (CE) to improve EBM training, and Recurrent Update to effectively update the CE and ERo jointly, enabling the graph encoder and energy head promote each other. Furthermore, we found that existing node OOD detection methods are evaluated only on homophilic graphs, with heterophilic graphs being overlooked. We therefore conduct a comprehensive evaluation of existing methods across homophilic and heterophilic graphs. The results show that existing graph-based methods deliver even worse performance on heterophilic graphs, compared to those graph-agnostic node OOD detection methods (Fig. 1). Thanks to the powerful constructed DeGEM, our method can achieve state-of-the-art performance on OOD detection across both homophilic and heterophilic graphs without OOD exposure.

Our key contributions can be summarized as follows:

- Our proposed DeGEM decomposes EBM into two components, a graph encoder for extracting topology information and an energy head for estimating density, which avoids the notorious challenges of sampling the adjacency matrix when training via MLE and prevents serious performance degradation on heterophilic graphs.
- We are the first to evaluate node OOD detection performance on both homophilic and heterophilic graphs and provide a comprehensive assessment of existing graph-based methods.

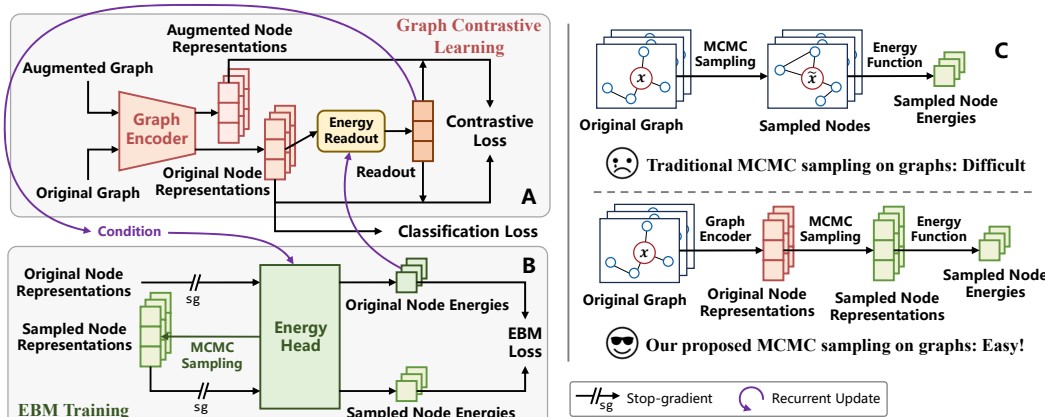

Figure 2: **A & B** The detailed graph contrastive learning and EBM training process. The readout of original node representations in **A** participates in the Conditional Energy (CE) in **B**, and the original node energies in **B** are delivered to **A** for Energy Readout (ERo). We propose a Recurrent Update mechanism to jointly train the CE and ERo effectively. **C** The comparison between traditional MCMC sampling and our proposed MCMC sampling.

- We conduct extensive experiments to demonstrate the superiority of DeGEM. The results show that DeGEM, trained without OOD exposure, outperforms state-of-the-art methods—whether trained with or without OOD exposure—on both homophilic and heterophilic graphs.

## 2 PRELIMINARY

**Notations.** We denote an undirected graph without self-loops as $\mathcal{G} = \{\mathbf{X}, \mathbf{A}\}$, where $\mathbf{X} = \{\boldsymbol{x}_i\}_{i=1}^N \in \mathbb{R}^{N \times d_0}$ is the initial node feature matrix, $d_0$ is the feature dimension, and $\mathbf{A} \in \mathbb{R}^{N \times N}$ is the adjacency matrix. $\mathcal{N}(\boldsymbol{x}_i)$ is the feature set of neighbors of node $\boldsymbol{x}_i$. $\mathbf{D}$ is a diagonal matrix standing for the degree matrix such that $\mathbf{D}_{ii} = \sum_{j=1}^N \mathbf{A}_{ij}$. $\mathbf{H}^{(\ell)} = \{\boldsymbol{h}_i^{(\ell)}\}_{i=1}^N \in \mathbb{R}^{N \times d}$ is the representation matrix in the $\ell$-th layer, where $d$ is the hidden dimension. We use $\mathbf{Y} = \{\boldsymbol{y}_i\}_{i=1}^N \in \mathbb{R}^{N \times C}$ to denote the ground-truth node label matrix, where $C$ is the number of classes and $\boldsymbol{y}_i$ is a one-hot vector.

**Graph Out-of-Distribution Node Detection.** OOD detection refers to identifying data samples that do not conform to the distribution of the training data, while keeping the classification capability of in-distribution (ID) data. Formally, an OOD detection score function $\mathcal{S}(\cdot, \cdot)$ should be defined to map the node and its neighbors to a scalar score, such that $\mathcal{S}(\boldsymbol{x}, \mathcal{N}(\boldsymbol{x}))$ yields a higher value for OOD nodes than for ID nodes. It can be seen that in the context of graph data, OOD detection becomes challenging due to the complex interplay between node features and graph topology. The OOD score for each node depends on itself and also the relational context provided by other nodes within the graph, which is distinct from OOD detection in the vision domain.

**Heterophily Issue.** Most traditional GNNs are designed based on the homophily assumption (Kipf & Welling, 2016; Velickovic et al., 2018), where linked nodes tend to be similar. However, heterophilic graphs—where linked nodes often belong to different categories—are prevalent in real-world applications, on which the traditional GCNs perform poorly (Pei et al., 2020). While extensive works have been proposed to address the heterophily issue (Abu-El-Haija et al., 2019; Bo et al., 2021; Zhu et al., 2020a; Chien et al., 2021; Li et al., 2022a; Chen et al., 2023; Luo et al., 2024a) in node classification tasks, it has not received sufficient attention in the context of node OOD detection, resulting in performance degradation when applying models designed for node OOD detection to heterophily graphs.

**Energy-based Model.** A deep EBM models (Du & Mordatch, 2019) the data distribution $p_d(\boldsymbol{x})$ using Boltzmann distribution $p_{\boldsymbol{\theta}}(\boldsymbol{x}) = \frac{\exp(-E_{\boldsymbol{\theta}}(\boldsymbol{x}))}{Z_{\boldsymbol{\theta}}}$ with the energy function $E_{\boldsymbol{\theta}}(\boldsymbol{x})$, where $Z_{\boldsymbol{\theta}}$ is the corresponding normalizing constant. EBM is trained by minimizing the negative log-likelihood (NLL) $\mathcal{L}_{\boldsymbol{\theta}}$ of $p_{\boldsymbol{\theta}}(\boldsymbol{x})$, such that $\mathcal{L}_{\boldsymbol{\theta}} = -\mathbb{E}_{\boldsymbol{x} \sim p_d(\boldsymbol{x})} \left[ \log p_{\boldsymbol{\theta}}(\boldsymbol{x}) \right]$. The EBM loss can be reformulated as follows:

$$\mathcal{L}_{\boldsymbol{\theta}} = \mathbb{E}_{\boldsymbol{x} \sim p_d(\boldsymbol{x})}\Big[ E_{\boldsymbol{\theta}}(\boldsymbol{x}) \Big] - \mathbb{E}_{\tilde{\boldsymbol{x}} \sim p_{\boldsymbol{\theta}}(\tilde{\boldsymbol{x}})}\Big[ E_{\boldsymbol{\theta}}(\text{sg}[\tilde{\boldsymbol{x}}]) \Big] \approx \frac{1}{N}\sum_{i=1}^{N} E_{\boldsymbol{\theta}}(\boldsymbol{x}_i) - \frac{1}{M}\sum_{j=1}^{M} E_{\boldsymbol{\theta}}(\text{sg}[\tilde{\boldsymbol{x}}_j]), \quad (1)$$

where $\text{sg}[\cdot]$ denotes the stop-gradient operation. Sampling $\tilde{\boldsymbol{x}}$ from $p_{\boldsymbol{\theta}}(\tilde{\boldsymbol{x}})$ can be achieved by a $K$-step Markov chain Monte Carlo (MCMC) sampling:

$$\tilde{\boldsymbol{x}}^{(k)} = \tilde{\boldsymbol{x}}^{(k-1)} - \lambda \nabla_{\tilde{\boldsymbol{x}}} E_{\boldsymbol{\theta}}(\tilde{\boldsymbol{x}}^{(k-1)}) + \boldsymbol{\epsilon}^{(k)},$$

where $\tilde{\boldsymbol{x}}^{(0)}$ is given initial samples, $\lambda$ is the step size, and $\boldsymbol{\epsilon}^{(k)} \sim \mathcal{N}(0, \sigma^2)$ with a given noise $\sigma^2$. More details of derivation can be found in the Appendix D.

## 3 METHODOLOGY

### 3.1 REVISITING GRAPH EBM FOR OOD DETECTION

Different from image data, the graph data is neither continuous nor i.i.d. The inherent challenge is how to define and train an EBM over $(\boldsymbol{x}, \mathcal{N}(\boldsymbol{x}))$. Since $\mathcal{N}(\boldsymbol{x})$ is discrete, potentially huge, and includes arbitrary neighbor nodes, it is hard to sample $\mathcal{N}(\boldsymbol{x})$ by MCMC for training EBM.

To avoid such challenge, GNNSafe (Wu et al., 2023b) proposes to train the EBMs by maximizing the conditional likelihood $\log p_{\boldsymbol{\theta}}(y|\boldsymbol{x}) = \frac{\exp(-E_{\boldsymbol{\theta}}(\boldsymbol{x},y))}{\sum_{y'}\exp(-E_{\boldsymbol{\theta}}(\boldsymbol{x},y'))}$ via node classification, where $E_{\boldsymbol{\theta}}(\boldsymbol{x},y) = -f_{\boldsymbol{\theta}}(\boldsymbol{x}, \mathcal{N}(\boldsymbol{x}))_{[y]}$ and $f_{\boldsymbol{\theta}}(\cdot, \cdot)$ is a classifier. Finally, the energy score of each node can be regarded as OOD score, i.e., $\mathcal{S}(\boldsymbol{x}, \mathcal{N}(\boldsymbol{x})) = E_{\boldsymbol{\theta}}(\boldsymbol{x}) = -\log\sum_{y}\exp(-E_{\boldsymbol{\theta}}(\boldsymbol{x},y))$.

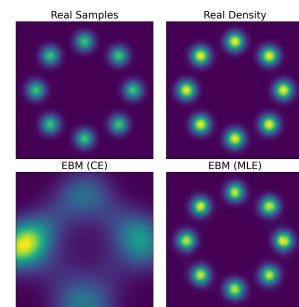

Figure 3: Visualization comparison on 2D data.

The EBMs trained in this way exhibit limitations in terms of capability and a heavy reliance on sufficient labeled data. In particular, it has not been trained for modeling the marginal data distribution, resulting in an inferior ability to capture the data distribution. For instance, it may falter even when tasked with handling a simple 2D dataset, i.e., 8 Gaussians (Fig. 3). This indicates that the energy constructed by the classification logits does not effectively capture the underlying data distribution. Consequently, such methods intuitively face challenges in addressing OOD detection for graph data, particularly when labeled data are scarce.

### 3.2 OVERVIEW OF OUR METHOD: DEGEM

The framework of our proposed method is shown in Fig. 2. DeGEM first utilizes a graph encoder trained by GCL algorithm (Veličković et al., 2018) to extract node representations $\boldsymbol{h} = g_{\boldsymbol{\alpha}}(\boldsymbol{x}, \mathcal{N}(\boldsymbol{x})) \in \mathbb{R}^d$. In this way, the topology information of the original graph is well-encoded into $\boldsymbol{h}$, such that the follow-up steps can be free from $\mathbf{A}$. Next, a $K$-step MCMC sampling is applied over the low-dimensional latent space $\tilde{\boldsymbol{h}} \sim q_{\boldsymbol{\omega}}(\tilde{\boldsymbol{h}})$ to learn an energy function $f_{\boldsymbol{\omega}}$, which is defined as an MLP. Please see Appendix E.1 for detailed training algorithm.

### 3.3 OUR DESIGN

The powerful capability of EBM comes from its minimal restrictions on modeling, but at the same time, this is also a double-edged sword that leads to difficult learning. To address the difficulty of defining EBMs on graphs, we propose to restrict EBM modeling formulation to some extent. Specifically, we suggest decomposing EBM into two parts: the first part focuses on extracting graph structural information, and the second part is dedicated to learning the energy.

For node $\boldsymbol{x}$ and its neighbor $\mathcal{N}(\boldsymbol{x})$, we can define the EBM as $E_{\boldsymbol{\theta}}(\boldsymbol{x}) = f_{\boldsymbol{\omega}} \circ g_{\boldsymbol{\alpha}}(\boldsymbol{x}, \mathcal{N}(\boldsymbol{x}))$, where $g_{\boldsymbol{\alpha}}(\boldsymbol{x}, \mathcal{N}(\boldsymbol{x})) = \boldsymbol{h} \in \mathbb{R}^d$ is a graph encoder that focuses on extracting the graph structural information, and $f_{\boldsymbol{\omega}}(\boldsymbol{h}) \in \mathbb{R}$ is an energy function that outputs the energy score.

Suppose we have a fixed $g_{\boldsymbol{\alpha}}$ that learns graph structural information well, following Eq. (1), the EBM loss for learning $f_{\boldsymbol{\omega}}$ can be reformulated as:

$$\mathcal{L}_{\text{ebm}} = \mathbb{E}_{p_d(\boldsymbol{x}, \mathcal{N}(\boldsymbol{x}))}\Big[ f_{\boldsymbol{\omega}}(\text{sg}[g_{\boldsymbol{\alpha}}(\boldsymbol{x}_i, \mathcal{N}(\boldsymbol{x}))]) \Big] - \mathbb{E}_{p_{\boldsymbol{\theta}}(\boldsymbol{x}, \mathcal{N}(\boldsymbol{x}))}\Big[ f_{\boldsymbol{\omega}}(\text{sg}[g_{\boldsymbol{\alpha}}(\boldsymbol{x}_i, \mathcal{N}(\boldsymbol{x}))]) \Big]$$

$$= \mathbb{E}_{p_{\boldsymbol{\alpha}}(\boldsymbol{h})}\Big[ f_{\boldsymbol{\omega}}(\text{sg}[\boldsymbol{h}]) \Big] - \mathbb{E}_{q_{\boldsymbol{\theta}}(\boldsymbol{h})}\Big[ f_{\boldsymbol{\omega}}(\text{sg}[\boldsymbol{h}]) \Big] \iff \text{KL}\Big( p_{\boldsymbol{\alpha}}(\boldsymbol{h}) \| q_{\boldsymbol{\theta}}(\boldsymbol{h}) \Big),$$

where $p_{\boldsymbol{\alpha}}(\boldsymbol{h}) = \int p_d(\boldsymbol{x})\delta(\boldsymbol{h} - g_{\boldsymbol{\alpha}}(\boldsymbol{x}))d\boldsymbol{x}$ and $q_{\boldsymbol{\theta}}(\boldsymbol{h}) = \int p_{\boldsymbol{\theta}}(\boldsymbol{x})\delta(\boldsymbol{h} - g_{\boldsymbol{\alpha}}(\boldsymbol{x}))d\boldsymbol{x}$ are the aggregated distribution over latent space of real data and EBM samples, respectively.

It can be observed that the optimal solution is achieved at $p_{\boldsymbol{\alpha}}(\boldsymbol{h}) = q_{\boldsymbol{\theta}}(\boldsymbol{h})$, and optimizing the objective actually equals minimizing the KL divergence between two distributions over latent space. Hence we can formulate a surrogate optimization objective for directly learning a latent EBM $f_{\boldsymbol{\omega}}$:

$$\mathrm{KL}\Big(p_{\boldsymbol{\alpha}}(\boldsymbol{h})\|q_{\boldsymbol{\omega}}(\boldsymbol{h})\Big) \iff \mathbb{E}_{p_{\boldsymbol{\alpha}}(\boldsymbol{h})}\Big[f_{\boldsymbol{\omega}}(\boldsymbol{h})\Big] - \mathbb{E}_{q_{\boldsymbol{\omega}}(\boldsymbol{h})}\Big[f_{\boldsymbol{\omega}}(\boldsymbol{h})\Big], \tag{2}$$

where $q_{\boldsymbol{\omega}}(\boldsymbol{h}) = \frac{\exp(-f_{\boldsymbol{\omega}}(\boldsymbol{h}))}{Z_{\boldsymbol{\omega}}}$. Sampling from $q_{\boldsymbol{\omega}}(\boldsymbol{h})$ can be achieved by running $K$-step MCMC sampling. By the surrogate objective, we can move the MCMC sampling from $(\boldsymbol{x}, \mathcal{N}(\boldsymbol{x}))$ space to the informative latent space, avoiding the awkward and challenging scenarios of needing to sample nodes under high-dimension and their neighbors.

A left challenge is obtaining a flexible $g_{\boldsymbol{\alpha}}$ to extract graph topology information. We suggest utilizing Graph Contrastive Learning (GCL) algorithm for training the graph encoder to extract node representations. This can not only extract informative features but also deal with the limited labeled data case. Among graph contrastive learning methods, we propose using DGI (Veličković et al., 2018) as the learning algorithm for $g_{\boldsymbol{\alpha}}$. Intuitively, DGI contrasts between nodes and the whole graph, while the OOD detection task requires detecting the outliner nodes in a whole graph, the contrast between the local features and the global can bring more benefits. The effect of different GCL algorithms will be shown in Sec. 4.4.

DGI first shuffles the rows of node features matrix to get an augmented graph $\{\mathbf{X}', \mathbf{A}'\}$, then obtains node representations by $\boldsymbol{h}_i = g_{\boldsymbol{\alpha}}(\boldsymbol{x}_i, \mathcal{N}(\boldsymbol{x}_i))$ and $\boldsymbol{h}'_j = g_{\boldsymbol{\alpha}}(\boldsymbol{x}'_j, \mathcal{N}(\boldsymbol{x}'_j))$, respectively. The contrastive loss for learning $g_{\boldsymbol{\alpha}}$ is given by:

$$\mathcal{L}_{\mathrm{cl}}(\boldsymbol{h}, \boldsymbol{s}) = -\frac{1}{2N}\left(\sum_{i=1}^{N}\log D_{\boldsymbol{\psi}}(\boldsymbol{h}_i, \boldsymbol{s}) + \sum_{j=1}^{N}\log(1 - D_{\boldsymbol{\psi}}(\boldsymbol{h}'_j, \boldsymbol{s}))\right), \tag{3}$$

where $\boldsymbol{s} = \frac{1}{N}\sum_{i=1}^{N}\boldsymbol{h}_i \in \mathbb{R}^d$ is the summary readout of all the node representations, which can be regarded as a global view of the whole graph. Following DGI (Veličković et al., 2018), we use a bilinear scoring function for the discriminator $D_{\boldsymbol{\psi}}(\cdot, \cdot)$, which is also learned by minimizing $\mathcal{L}_{\mathrm{cl}}$.

Additionally, the node representations $\boldsymbol{h}_i$ can be further enhanced by ID nodes classification with a prediction head $I_{\boldsymbol{\phi}}$ (e.g., MLP) as $\hat{\boldsymbol{y}}_i = I_{\boldsymbol{\phi}}(\boldsymbol{h}_i)$, and the cross-entropy loss $\mathcal{L}_{\mathrm{cls}}$ is used for learning the prediction head $I_{\boldsymbol{\phi}}(\cdot)$ and also graph encoder $g_{\boldsymbol{\alpha}}(\cdot, \cdot)$.

**Multi-Hop (MH) Graph Feature Encoder.** Simply using a one-layer GCN, as in DGI (Veličković et al., 2018), cannot fully extract informative representations from original graph features. Additionally, similar nodes often lie in the distance in heterophilic graphs, so combining long-range information could bring positive effects in addressing heterophily issue. Therefore, a $L$-layer GNN is employed to learn graph node features in our proposed method. Furthermore, the weighted self-loops technique is used as a simple enhanced method for graph filter (Bo et al., 2021; Chen et al., 2023). Finally, all the intermediate features (the output of all the layers) are fused to obtain the final representations.

First, a symmetric normalized adjacency with weighted self-loops is used for extracting node features:

$$\mathbf{X}^{(\ell)} = \left(\beta\mathbf{I} + \mathbf{D}^{-\frac{1}{2}}\mathbf{A}\mathbf{D}^{-\frac{1}{2}}\right)\mathbf{X}^{(\ell-1)}, \quad \mathbf{X}^{(0)} = \mathbf{X}, \tag{4}$$

where $\beta$ is a hyper-parameter. Note that this operation for the positive samples can be done in preprocessing since it does not involve any learnable parameters, which greatly reduce the running time. Then the intermediate features are transformed and fused to obtain the final representations:

$$\mathbf{H} = \left[\mathbf{H}^{(0)}\|\mathbf{H}^{(1)}\|\cdots\|\mathbf{H}^{(L)}\right]\mathbf{W}_{\mathrm{enc}} + \boldsymbol{b}_{\mathrm{enc}}, \quad \mathbf{H}^{(\ell)} = \mathbf{X}^{(\ell)}\mathbf{W}^{(\ell)} + \boldsymbol{b}^{(\ell)}, \tag{5}$$

where $\mathbf{W}$ and $\boldsymbol{b}$ are learnable weights and bias. $\|$ is a concatenation operation. The combination of Eqs. (4) and (5) is denoted by $\mathbf{H} = g_{\boldsymbol{\alpha}}(\mathbf{X}, \mathbf{A})$.

**Conditional Energy (CE).** The readout summary $\boldsymbol{s}$ of ID dataset can be used as supervised information in energy function $f_{\boldsymbol{\omega}}$, to compare between node representations and global representations

when computing energy. For instance, $f_{\boldsymbol{\omega}}(\boldsymbol{h}_i, \boldsymbol{s}) = \mathbf{W}[\boldsymbol{h}_i \| \rho \boldsymbol{s}] + \boldsymbol{b}$, $f_{\boldsymbol{\omega}}(\boldsymbol{h}_i, \boldsymbol{s}) = \boldsymbol{h}_i^\top \mathbf{W} \boldsymbol{s}$, and etc., where $\rho$ is a hyper-parameter, $\mathbf{W}$ and $\boldsymbol{b}$ are learned parameters. The benefit of the global information introduced by the proposed CE is twofold: 1) It can guide MCMC sampling, reducing the sampling difficulty; 2) It can act as global prior information when detecting OOD nodes.

**Energy Readout (ERo).** In DGI, the readout summary $\boldsymbol{s}$ is obtained by simply averaging all the node representations $\boldsymbol{s} = \frac{1}{N} \sum_{i=1}^{N} \boldsymbol{h}_i$. However, the importance of each node may not be the same. Considering that the density of a node can represent the importance of the node to some extent and energy is un-normalized density, we propose to use energy for conducting the weighted readout:

$$\boldsymbol{s}_p = \gamma \sum_{i=1}^{N} \bar{p}_i \boldsymbol{h}_i + (1 - \gamma) \frac{1}{N} \sum_{i=1}^{N} \boldsymbol{h}_i = \sum_{i=1}^{N} \left( \gamma \bar{p}_i + \frac{1 - \gamma}{N} \right) \boldsymbol{h}_i,$$

where $\bar{p}_i = \mathrm{softmax}(-f_{\boldsymbol{\omega}}(\boldsymbol{h}_i)) = \frac{\exp(-f_{\boldsymbol{\omega}}(\boldsymbol{h}_i))}{\sum_{j=1}^{N} \exp(-f_{\boldsymbol{\omega}}(\boldsymbol{h}_j))} \in \mathbb{R}$ is the weight, and $\gamma$ is a hyper-parameter for a trade-off between weight average and direct average.

## 3.4 Recurrent Update For Learning DeGEM

To apply CE and ERo simultaneously, the energy readout $\boldsymbol{s}_p$ and conditional energy $f_{\boldsymbol{\omega}}(\boldsymbol{h}_i, \boldsymbol{s})$ should be updated recurrently. The Recurrent Update can make energy function $f_{\boldsymbol{\omega}}$ and graph encoder $g_{\boldsymbol{\alpha}}$ promote each other, leading to better learning. Next, we introduce the Recurrent Update in details. The weights of conditional energy are initialized as uniform distribution $\bar{\boldsymbol{p}}^{(0)} = \left[ \frac{1}{N}, \frac{1}{N}, \cdots, \frac{1}{N} \right] \in \mathbb{R}^N$. In the $e$-th epoch, node representations of original graph $\{\mathbf{X}, \mathbf{A}\}$ and shuffled graph $\{\mathbf{X}', \mathbf{A}'\}$ are first obtained by $\boldsymbol{h}_i = g_{\boldsymbol{\alpha}}(\boldsymbol{x}_i, \mathcal{N}(\boldsymbol{x}_i))$ and $\boldsymbol{h}_j' = g_{\boldsymbol{\alpha}}(\boldsymbol{x}_j', \mathcal{N}(\boldsymbol{x}_j'))$, respectively, and $\tilde{\boldsymbol{h}}_u \sim q_{\boldsymbol{\omega}}(\tilde{\boldsymbol{h}})$ is sampled by $K$-step MCMC sampling, then the final learning at each iteration is conducted as follows:

**Step 1: Learning energy head $f_{\boldsymbol{\omega}}$ given graph encoder $g_{\boldsymbol{\alpha}}$:** The energy head $f_{\boldsymbol{\omega}}$ is learned by MLE over informative latent space by a surrogate objective given the well-trained graph encoder $g_{\boldsymbol{\alpha}}$ and according global representation $\bar{\boldsymbol{s}}_p^{(e)}$:

$$\bar{\boldsymbol{s}}_p^{(e)} = \sum_{i=1}^{N} \left( \gamma \bar{p}_i^{(e-1)} + \frac{1 - \gamma}{N} \right) \mathrm{sg}[\boldsymbol{h}_i],$$

$$\mathcal{L}_{\boldsymbol{\omega}} = \mathcal{L}_{\mathrm{ebm}} = \frac{1}{N} \sum_{i=1}^{N} f_{\boldsymbol{\omega}}(\mathrm{sg}[\boldsymbol{h}_i], \bar{\boldsymbol{s}}_p^{(e)}) - \frac{1}{M} \sum_{u=1}^{M} f_{\boldsymbol{\omega}}(\mathrm{sg}[\tilde{\boldsymbol{h}}_u], \bar{\boldsymbol{s}}_p^{(e)}).$$

**Step 2: Learning graph encoder $g_{\boldsymbol{\alpha}}$ given energy head $f_{\boldsymbol{\omega}}$:** The Graph Encoder $g_{\boldsymbol{\alpha}}$ is learned by the mix of graph contrastive learning and node classification given well-trained energy head $f_{\boldsymbol{\omega}}$ for weighted readout as follows:

$$\bar{p}_i^{(e)} = \mathrm{softmax}\left( -f_{\boldsymbol{\omega}}(\mathrm{sg}[\boldsymbol{h}_i], \bar{\boldsymbol{s}}_p^{(e)}) \right),$$

$$\boldsymbol{s}_p^{(e)} = \sum_{i=1}^{N} \left( \gamma \bar{p}_i^{(e)} + \frac{1 - \gamma}{N} \right) \boldsymbol{h}_i,$$

$$\mathcal{L}_{\boldsymbol{\alpha}, \boldsymbol{\psi}, \boldsymbol{\phi}} = \mathcal{L}_{\mathrm{cl}}(\boldsymbol{h}, \boldsymbol{s}_p^{(e)}) + \xi \mathcal{L}_{\mathrm{cls}}(I_{\boldsymbol{\phi}}(\boldsymbol{h}), \boldsymbol{y}),$$

where $\xi$ is a hyper-parameter, $\mathcal{L}_{\mathrm{cls}}$ is the cross-entropy loss. See Algorithm 1 for the detailed learning process.

**Inference.** For an unknown node $\boldsymbol{x}_v$, its representation is obtained by $\boldsymbol{h}_v = g_{\boldsymbol{\alpha}}(\boldsymbol{x}_v, \mathcal{N}(\boldsymbol{x}_v))$, then the energy score $f_{\boldsymbol{\omega}}(\boldsymbol{h}_v, \boldsymbol{s}^{(E)})$ serves as the OOD score, where $\boldsymbol{s}^{(E)}$ is the summary readout of ID dataset in the final training epoch, and its label prediction is computed by $\hat{\boldsymbol{y}}_v = I_{\boldsymbol{\phi}}(\boldsymbol{h}_v)$.

## 3.5 Understanding of Using DGI for Learning Graph Encoder

By comparing Eq. (3) and Eq. (2), it can be found that there are some similarities between these two objectives. Actually, the learning of DGI can be understood as learning an EBM with global condition by noise contrastive estimation (Aneja et al., 2021; Gutmann & Hyvärinen, 2010) as the following form:

Table 1: OOD detection performance on four **homophilic datasets** (`Cora`, `Amazon`, `Twitch` and `Arxiv`) measured by AUC (↑) and Acc (↑) with three OOD types (Structure manipulation, Feature interpolation, Label leave-out). Other results for AUPR, FPR95 are deferred to Appendix I.

| Method | Cora | | | | | | Amazon-Photo | | | | | | Twitch | | | | Arxiv | | | | Avg | |
| | Structure | | Feature | | Label | | Structure | | Feature | | Label | | ES | FR | RU | | 2018 | 2019 | 2020 | | | |
| | AUC↑ | Acc↑ | AUC↑ | Acc↑ | AUC↑ | Acc↑ | AUC↑ | Acc↑ | AUC↑ | Acc↑ | AUC↑ | Acc↑ | AUC↑ | AUC↑ | AUC↑ | Acc↑ | AUC↑ | AUC↑ | AUC↑ | Acc↑ | AUC↑ | Acc↑ |
|---|---|---|---|---|---|---|---|---|---|---|---|---|---|---|---|---|---|---|---|---|---|---|
| MSP | 75.05 | 77.30 | 85.87 | 77.30 | 91.10 | 88.29 | 98.94 | 93.64 | 97.02 | 92.89 | 95.49 | 95.84 | 66.08 | 47.40 | 57.41 | 63.89 | 61.76 | 63.52 | 67.06 | 53.87 | 83.07 | 80.38 |
| ODIN | 30.57 | 74.60 | 21.19 | 77.50 | 20.29 | 87.66 | 3.50 | 91.42 | 5.31 | 92.91 | 10.16 | 96.08 | 43.97 | 51.84 | 49.53 | 62.05 | 44.01 | 42.63 | 38.96 | 49.29 | 22.67 | 78.94 |
| Mahalanobis | 41.03 | 71.90 | 63.92 | 74.20 | 67.45 | 88.92 | 62.40 | 93.42 | 72.47 | 92.88 | 60.80 | 95.84 | 46.73 | 49.69 | 38.38 | 62.95 | 57.08 | 56.76 | 56.92 | 51.59 | 58.74 | 78.96 |
| Energy | 79.48 | 79.10 | 89.34 | 79.30 | 93.26 | 90.19 | 99.94 | 93.04 | 98.51 | 92.76 | 97.13 | 95.68 | 58.42 | 72.91 | 69.90 | 65.59 | 64.61 | 65.90 | 70.37 | 53.92 | 86.46 | 81.20 |
| ResidualFlow | 61.23 | 82.20 | 60.72 | 82.20 | 64.69 | 82.20 | 82.58 | 93.55 | 75.46 | 93.55 | 78.31 | 93.55 | 62.60 | 55.59 | 67.68 | 66.37 | 62.38 | 63.82 | 62.21 | 54.84 | 68.47 | 81.06 |
| GKDE | 84.49 | 78.00 | 90.75 | 81.70 | 94.59 | 91.77 | 92.68 | 89.69 | 54.52 | 31.90 | 76.35 | 88.55 | 57.52 | 57.48 | 46.25 | 60.42 | 69.44 | 71.32 | 71.73 | 21.15 | 77.25 | 67.52 |
| GPN | 82.21 | 81.00 | 88.06 | 78.80 | 91.74 | 91.77 | 90.35 | 82.63 | 86.47 | 63.38 | 89.90 | 89.30 | 84.07 | 76.32 | 78.36 | 60.29 | OOM | OOM | OOM | OOM | N/A | N/A |
| OODGAT | 53.75 | 34.90 | 57.03 | 14.50 | 95.57 | 89.56 | 71.41 | 25.42 | 70.93 | 25.21 | 99.18 | 96.16 | 77.35 | 77.72 | 73.24 | 60.29 | 72.35 | 73.97 | 72.30 | 54.69 | 74.61 | 50.09 |
| GNNSafe | 87.98 | 75.30 | 92.18 | 75.40 | 92.36 | 88.92 | 98.69 | 93.74 | 98.47 | 92.96 | 97.34 | 95.72 | 51.00 | 79.08 | 82.93 | 66.18 | 67.27 | 69.20 | 79.02 | 54.26 | 88.73 | 80.31 |
| DeGEM (Ours) | 99.93 | 84.20 | 99.84 | 84.30 | 97.58 | 93.04 | 100.00 | 94.49 | 99.91 | 93.97 | 99.28 | 95.80 | 94.83 | 97.36 | 94.76 | 64.51 | 81.30 | 86.00 | 86.01 | 58.20 | 97.08 | 83.56 |

$$\mathcal{L}_{\mathrm{cl}} = \mathbb{E}_{p_d(\boldsymbol{x},\mathcal{N}(\boldsymbol{x}))}\Big[r(\boldsymbol{x},\mathcal{N}(\boldsymbol{x})|\boldsymbol{s})\Big] - \mathbb{E}_{q(\boldsymbol{x},\mathcal{N}(\boldsymbol{x}))}\Big[r(\boldsymbol{x},\mathcal{N}(\boldsymbol{x})|\boldsymbol{s})\Big],$$

where $q(\boldsymbol{x},\mathcal{N}(\boldsymbol{x})) = \int q(\boldsymbol{x},\mathcal{N}(\boldsymbol{x})|\tilde{\boldsymbol{x}},\mathcal{N}(\tilde{\boldsymbol{x}}),\boldsymbol{s})p_d(\tilde{\boldsymbol{x}},\mathcal{N}(\tilde{\boldsymbol{x}}))d\tilde{\boldsymbol{x}}$, $q(\boldsymbol{x},\mathcal{N}(\boldsymbol{x})|\tilde{\boldsymbol{x}},\mathcal{N}(\tilde{\boldsymbol{x}}),\boldsymbol{s})$ denotes the density of augmented samples given data samples, and $r(\boldsymbol{x},\mathcal{N}(\boldsymbol{x})|\boldsymbol{s}) = D_\psi(g_\alpha(\boldsymbol{x},\mathcal{N}(\boldsymbol{x})),\boldsymbol{s})$. After perfectly training, the optimal $r$ has following form $r^*(\boldsymbol{x},\mathcal{N}(\boldsymbol{x})|\boldsymbol{s}) = \frac{p_d(\boldsymbol{x},\mathcal{N}(\boldsymbol{x})|\boldsymbol{s})}{p_d(\boldsymbol{x},\mathcal{N}(\boldsymbol{x})|\boldsymbol{s})+q(\boldsymbol{x},\mathcal{N}(\boldsymbol{x})|\boldsymbol{s})}$. The inherent connection between DGI and EBM enables the effectiveness of constructing energy function by the learned $g_\alpha$. However, the optimal DGI learns the density ratio between data distribution and randomly augmented data distribution, which is unsuitable for OOD detection, hence it still needs to learn $f_\omega$ for output energy.

## 4 EXPERIMENTS

### 4.1 SETUP

**Dataset and Evaluation Metrics.** We evaluate DeGEM on seven benchmark datasets for node classification tasks (Yang et al., 2016; Shchur et al., 2018; Rozemberczki et al., 2021; Wang et al., 2020; Pei et al., 2020), including four *homophily* datasets (`Cora`, `Amazon-Photo`, `Twitch`, and `ogbn-Arxiv`) and three *heterophily* datasets (`Chameleon`, `Actor`, and `Cornell`). We mainly follow (Wu et al., 2021; 2023b) to adopt two established methods to simulate OOD scenarios. In the multi-graph context, OOD samples stem from distinct graphs or subgraphs not linked to the training nodes. Conversely, in the single-graph setting, OOD samples are part of the same graph as the training data but remain unseen during training. For `Twitch`, we treat one subgraph as in-distribution (ID) and others as OOD, using one for OOD exposure during training. In `ogbn-Arxiv`, we split the nodes by publication year for ID, OOD, and OOD exposure sets. For `Cora`, `Amazon`, `Chameleon`, `Actor`, and `Cornell` that have no clear domain information, we synthesize OOD data in three ways: i) Structure manipulation (S); ii) Feature interpolation (F); iii) Label leave-out (L). See detailed settings and splits in Appendix E.2. For the assessment of OOD detection performance, we employ standard metrics: Area Under the Receiver Operating Characteristic curve (AUC), Area Under the Precision-Recall curve (AUPR), and the False Positive Rate at 95% True Positive Rate (FPR95). In-distribution (ID) performance is quantified using the accuracy (Acc) metric on the testing nodes. In the following text, we use AUC to denote AUC with a little bit of abuse. Due to the space limits, we mainly present the results of AUC and Acc, the full results with AUPR and FPR95 can be found in Appendix I.

**Baseline Comparisons.** Our model is benchmarked against two categories of baseline methods. The first category comprises models that handle i.i.d. inputs, which are predominantly used in computer vision. These include MSP (Hendrycks & Gimpel, 2016), ODIN (Liang et al., 2018), Mahalanobis (Lee et al., 2018), OE (Hendrycks et al., 2018), and Energy (-FT) (Liu et al., 2020). We also include ResidualFlow (Zisselman & Tamar, 2020), a density-based method capable of modeling data distribution. For a fair comparison, we substitute the CNN backbones in these models with a GCN encoder. The second category consists of methods tailored for graph-structured data, such as GKDE (Zhao et al., 2020b), GPN (Stadler et al., 2021b), OODGAT (Song & Wang, 2022) and GNNSafe(++) (Wu et al., 2023a). It is noteworthy that OE, Energy-FT and GNNSafe++ incorporate OOD samples during the training phase, i.e., trained with OOD exposure.

**Implemetation Details.** We implement our model by PyTorch and conduct experiments on 24GB RTX-3090ti. Epoch number $E = 200$, MH layer number $L = 5$, hidden dimension $d = 512$, MCMC steps $K = 20$. We use Optuna (Akiba et al., 2019) to search hyper-parameters for our proposed model and baselines (see Appendix E.3 for detailed search space).

Table 2: OOD detection performance three **heterophilic datasets** (`Chameleon`, `Actor`, `Cornell`) measured by AUC (↑) and Acc (↑) on with three OOD types (Structure manipulation, Feature interpolation, Label leave-out). Other results for AUPR, FPR95 are deferred to Appendix I.

| Method | Chameleon | | | | | | Actor | | | | | | Cornell | | | | | | Avg | |
| | Structure | | Feature | | Label | | Structure | | Feature | | Label | | Structure | | Feature | | Label | | | |
| | AUC↑ | Acc↑ | AUC↑ | Acc↑ | AUC↑ | Acc↑ | AUC↑ | Acc↑ | AUC↑ | Acc↑ | AUC↑ | Acc↑ | AUC↑ | Acc↑ | AUC↑ | Acc↑ | AUC↑ | Acc↑ | AUC↑ | Acc↑ |
|---|---|---|---|---|---|---|---|---|---|---|---|---|---|---|---|---|---|---|---|---|
| MSP | 99.28 | 33.41 | 70.92 | 32.42 | 53.32 | 43.89 | 71.38 | 21.64 | 59.11 | 24.34 | 56.32 | 37.27 | 81.50 | 41.50 | 71.89 | 43.54 | 68.50 | 63.81 | 70.25 | 37.98 |
| ODIN | 60.09 | 28.80 | 53.20 | 33.24 | 68.07 | 37.01 | 42.32 | 22.30 | 52.57 | 23.90 | 65.10 | 35.79 | 67.99 | 38.78 | 85.76 | 38.10 | 42.16 | 63.81 | 59.69 | 35.75 |
| Mahalanobis | 99.59 | 29.90 | 58.93 | 33.41 | 42.42 | 44.54 | 79.96 | 24.23 | 65.94 | 23.72 | 52.58 | 31.98 | 69.84 | 36.73 | 70.08 | 43.54 | 81.73 | 62.86 | 69.01 | 36.77 |
| Energy | 91.94 | 37.14 | 67.75 | 37.58 | 59.92 | 41.69 | 64.26 | 25.21 | 51.71 | 24.33 | 55.00 | 36.71 | 83.09 | 38.10 | 86.70 | 38.10 | 69.70 | 62.86 | 70.01 | 37.97 |
| ResidualFlow | 48.24 | 34.50 | 53.82 | 34.50 | 56.16 | 34.50 | 49.72 | 27.63 | 50.07 | 27.63 | 50.68 | 27.63 | 67.80 | 43.54 | 66.33 | 43.54 | 73.09 | 43.54 | 57.32 | 35.22 |
| GKDE | 96.06 | 30.12 | 67.33 | 34.94 | 60.22 | 40.40 | 71.27 | 25.63 | 58.08 | 19.56 | 53.72 | 33.60 | 80.53 | 14.97 | 77.48 | 42.86 | 81.18 | 63.81 | 71.76 | 33.99 |
| GPN | 82.90 | 20.41 | 64.99 | 30.99 | 72.68 | 34.71 | 78.58 | 18.67 | 62.13 | 20.21 | 75.04 | 38.08 | 89.68 | 43.54 | 83.36 | 42.86 | 82.93 | 63.81 | 76.92 | 34.81 |
| OODGAT | 54.89 | 26.06 | 53.86 | 29.73 | 65.33 | 40.68 | 51.25 | 23.65 | 52.00 | 25.77 | 65.39 | 36.30 | 67.16 | 42.18 | 69.42 | 42.86 | 68.52 | 64.76 | 60.87 | 36.89 |
| GNNSafe | 34.36 | 35.33 | 57.46 | 38.07 | 52.18 | 43.43 | 31.76 | 26.30 | 50.66 | 26.20 | 51.60 | 37.92 | 74.66 | 25.17 | 76.22 | 41.50 | 68.17 | 63.81 | 55.23 | 37.52 |
| **DeGEM (Ours)** | **99.99** | **57.82** | **99.70** | **57.93** | **89.68** | **64.46** | **99.76** | **31.97** | **99.98** | **33.87** | **100.00** | **36.02** | **97.97** | **48.30** | **100.00** | **65.31** | **90.90** | **77.14** | **97.55** | **52.53** |

Table 3: OOD detection results on varying the available label ratio, measured by the average AUC↑ / AUPR↑ / FPR95↓.

| | OOD | Cora | | | Twitch | | | Chameleon | | | Cornell | | | Avg | | |
| | Expo | AUC↑ | AUPR↑ | FPR95↓ | AUC↑ | AUPR↑ | FPR95↓ | AUC↑ | AUPR↑ | FPR95↓ | AUC↑ | AUPR↑ | FPR95↓ | AUC↑ | AUPR↑ | FPR95↓ |
|---|---|---|---|---|---|---|---|---|---|---|---|---|---|---|---|---|
| Label Rate = 10% | | | | | | | | | | | | | | | | |
| OE | Yes | 65.36 | 47.01 | 88.69 | 63.36 | 71.92 | 91.87 | 67.29 | 74.66 | 89.12 | 58.06 | 65.42 | 94.78 | 63.52 | 64.76 | 91.12 |
| Energy-FT | Yes | 72.00 | 53.02 | 83.94 | 86.14 | 90.54 | 70.87 | 56.26 | 62.77 | 85.67 | 60.36 | 70.78 | 99.12 | 68.69 | 69.28 | 84.90 |
| GNNSafe++ | Yes | 78.74 | 64.55 | 79.40 | 77.75 | 85.01 | 84.30 | 33.21 | 47.70 | 93.77 | 69.52 | 79.51 | 94.34 | 64.80 | 69.19 | 87.95 |
| **DeGEM** | No | **97.87** | **94.79** | **10.00** | **94.67** | **97.47** | **51.41** | **96.64** | **98.27** | **14.14** | **97.57** | **98.95** | **11.25** | **96.69** | **97.37** | **21.70** |
| Label Rate = 50% | | | | | | | | | | | | | | | | |
| OE | Yes | 78.63 | 60.48 | 70.92 | 58.15 | 68.83 | 93.38 | 76.32 | 79.70 | 67.46 | 73.94 | 79.59 | 90.20 | 71.76 | 72.15 | 80.49 |
| Energy-FT | Yes | 77.58 | 56.54 | 65.56 | 87.30 | 90.72 | 53.09 | 68.60 | 74.14 | 88.59 | 67.91 | 75.24 | 94.37 | 75.35 | 74.16 | 75.40 |
| GNNSafe++ | Yes | 86.47 | 72.80 | 57.39 | 96.14 | 96.14 | 41.11 | 44.67 | 59.29 | 92.69 | 73.12 | 82.01 | 95.28 | 74.42 | 77.56 | 69.12 |
| **DeGEM** | No | **97.53** | **91.95** | **8.87** | **94.21** | **97.21** | **49.51** | **97.03** | **98.29** | **13.10** | **96.32** | **96.20** | **16.37** | **96.27** | **95.91** | **21.96** |
| Label Rate = 100% | | | | | | | | | | | | | | | | |
| OE | Yes | 83.35 | 68.91 | 67.19 | 62.38 | 73.53 | 88.43 | 81.07 | 82.83 | 52.23 | 71.71 | 72.47 | 74.05 | 74.63 | 74.44 | 70.48 |
| Energy-FT | Yes | 89.05 | 74.77 | 44.74 | 87.38 | 90.43 | 46.97 | 79.89 | 82.72 | 54.00 | 78.81 | 85.93 | 84.65 | 83.78 | 83.46 | 57.59 |
| GNNSafe++ | Yes | 92.69 | 84.62 | 40.63 | **96.58** | **98.03** | **24.76** | 48.23 | 63.37 | 93.67 | 79.10 | 85.10 | 82.65 | 79.15 | 82.78 | 60.43 |
| **DeGEM** | No | **99.16** | **97.78** | **4.25** | 95.65 | 97.93 | 29.11 | **96.46** | **98.04** | **13.38** | **96.29** | **97.61** | **23.75** | **96.89** | **97.84** | **17.62** |

## 4.2 EVALUATION RESULTS

We report the results of the proposed DeGEM and competing baselines on different datasets in Tabs. 1 and 2. Our finding indicates that the DeGEM consistently outperforms competing baselines without OOD exposure in terms of AUC and Acc across both homophilic and heterophilic datasets. Specifically, DeGEM increases the average AUC by 4.26% (resp. 20.63%) on homophilic (resp. heterophilic) graphs and improves the average Acc for homophilic (resp. heterophilic) graphs by 2.37% (resp. 13.95%). On the contrary, although the baselines tailored for graph inputs surpass those designed for i.i.d. data on homophilic graphs, they show a lower performance on heterophilic graphs. Additionally, after adding the Energy Propagation technique to Energy (i.e., GNNSafe), the average AUC decreases significantly from 70.01% to 55.23% and 56.42%, indicating that the Energy Propagation technique will cause severe performance decrease on heterophilic graphs.

The high performance of DeGEM is primarily attributed to the enhancement brought by the robust representation learned by the GCL algorithm and the classification loss, as well as the powerful data modeling capabilities of EBM trained via the MLE approach. Overall, these findings underscore the superiority of DeGEM over baseline approaches, on both homophilic and heterophilic graphs.

## 4.3 EVALUATION ON LIMITED LABELS

Given the time-consuming and labor-intensive nature of obtaining node labels on graphs in the real world, an interesting evaluation task is to evaluate the OOD detection performance of a model with limited category labels. We follow the data splits used in the experiments of Sec. 4.2 as a foundation and progressively reduce the proportion of available labels in the training set: from 100% to 50%, and then to 10%. To highlight the performance of our method, we select three baselines OE, Energy-FT

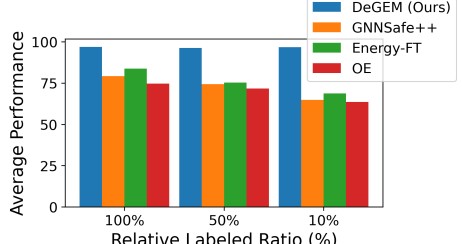

Figure 4: Performance across labeled ratios.

and GNNSafe++ which incorporate OOD samples during the training phase, i.e., trained with OOD exposure. Note that *we do not reduce the number of nodes used for OOD exposure training*, but only decrease the number of category labels available within the distribution. We present the average results of each dataset in Tab. 3, and the average performance of each method in Fig. 4 for more intuitive comparison. We report the full results in Appendix I. The results show that our proposed DeGEM maintains high performance under different label proportions, with no significant decline; in contrast, all baselines exhibit a marked performance drop as the labels ratio decrease. In particular, when the available label rate is 100%, our method still outperforms baselines, highlighting the

Table 4: Ablation study measured by the average AUC↑. 'Eprop' stands for energy propagation. 'MLE-Energy' indicates whether the energy head is trained via maximum likelihood estimation, otherwise, the node energy is obtained by classification logits (Classify-Energy). 'GCL' indicates whether a graph contrastive learning algorithm is employed when training the graph encoder.

| | Components | | | | | | | Homophily | | | Heterophily | | | Avg |
| | Eprop | Classify-Energy | MLE-Energy | GCL | MH | CE | ERo | Cora | Twitch | Avg | Cham. | Cornell | Avg | |
|---|---|---|---|---|---|---|---|---|---|---|---|---|---|---|
| Energy | | ✓ | | | | | | 87.36 | 67.07 | 77.22 | 73.20 | 79.83 | 76.51 | 76.87 |
| GNNSafe | ✓ | ✓ | | | | | | 90.84 | 71.00 | 80.92 | 48.00 | 73.02 | 60.51 | 70.72 |
| | | ✓ | | ✓ | | | | 83.13 | 47.61 | 65.37 | 82.33 | 82.54 | 82.44 | 73.90 |
| | | | ✓ | | | | | 69.11 | 74.43 | 71.77 | 77.51 | 67.91 | 72.71 | 72.24 |
| | | | ✓ | ✓ | | | | 88.87 | 84.77 | 86.82 | 90.63 | 87.00 | 88.82 | 87.82 |
| | | | ✓ | ✓ | | ✓ | | 88.82 | 69.81 | 79.32 | 92.57 | 84.95 | 88.76 | 84.04 |
| | | | ✓ | ✓ | | | ✓ | 92.34 | 83.12 | 87.73 | 93.57 | 86.31 | 89.94 | 88.84 |
| | | | ✓ | ✓ | | ✓ | ✓ | 92.80 | 84.77 | 88.78 | 94.46 | 85.35 | 89.91 | 89.35 |
| | | | ✓ | ✓ | ✓ | | | 97.47 | 93.61 | 95.54 | 95.23 | 95.90 | 95.57 | 95.55 |
| | | | ✓ | ✓ | ✓ | ✓ | | 98.03 | 94.96 | 96.50 | 96.24 | 95.49 | 95.87 | 96.18 |
| | | | ✓ | ✓ | ✓ | | ✓ | 96.24 | 92.41 | 94.32 | 95.80 | 93.99 | 94.89 | 94.61 |
| **DeGEM** | | | ✓ | ✓ | ✓ | ✓ | ✓ | **99.12** | **95.65** | **97.38** | **96.46** | **96.29** | **96.37** | **96.88** |

superiority of our method. These results demonstrate that our method, benefiting from its effective design, can consistently and effectively extract graph topology information and learn the data density well, thereby significantly outperforming previous state-of-the-art graph OOD detection methods in the more practical scenario of limited labels.

## 4.4 ABLATION STUDY

**The Impact of Proposed Techniques.** In what follows, we evaluate the impact of training the energy head via MLE (`MLE-Energy`) versus deriving node energy directly from classification logits (`Classify-Energy`). Additionally, we assess the use of DGI algorithm during graph encoder training (`GCL`), and explore the effectiveness of the Energy-Propagation technique (`Eprop`), the Multi-Hop Graph Feature Encoder (`MH`), Conditional Energy (`CE`), and Energy Readout (`ERo`). We present the results in Tab. 4. The baseline (Row 1) utilizes a GCN for node classification and obtains the node energy by the classification logits. Since `Eprop` is based on homophily assumption, the performance increases slightly (Avg +3.7%) on homophilic graphs but shows a dramatic degradation (Avg -16%) on heterophilic graphs (Row 2). Compared to the baseline (Avg 76.87%), solely utilizing the MLE for training the energy head (Avg 72.24%) or employing GCL algorithm for training the graph encoder (Avg 73.90%) does not work. It is the combination of MLE and GCL algorithm that brings positive effects for OOD detection tasks (Avg 87.82%). More detailed discussion can be found on **Observation 1–4** below. With `MH`, DeGEM achieves a significant performance boost (Avg +11%, Row 9), demonstrating that combining local and global information simultaneously enhances performance on both homophilic and heterophilic graphs. `CE` alone can improve the average performance (Avg +0.63%, Row 10); however, using `ERo` on its own does not yield similar benefits (Avg -0.94%, Row 11). This is because `CE` takes the global view into account, producing better node energy, which enables `ERo` to generate a more optimized readout summary. Therefore combining them together via `Recurrent Update` can bring more positive effects (Avg +1.33%, Row 12).

**Observation 1: combining DGI with classification does not work.** As shown in Tab. 4, although the `Classifier-Energy` variant shows 5.93% improvement of AUC on heterophilic graphs after adding DGI loss (`GCL`) into the classification task (Row 3), its performance on homophilic graphs dramatically decrease from 77.22% to 65.37%. This indicates that the integration of DGI algorithm and `Classifier-Energy` has an overall negative impact on its performance.

**Observation 2: naively training Energy Head via MLE does not work.** As shown in Tab. 4, when decoupling the model into two parts (a graph encoder and an energy head), if we train the energy head via MLE but only employ classification loss for training the graph encoder (Row 4), the model will show a worse performance compared to the `Classifier-Energy` variant (Row 1).

**Observation 3: combining DGI and MLE works well** As shown in Tab. 4, after adding DGI loss (`GCL`), the performance of `MLE-Energy` variant dramatically improves from 72.74% to 87.82% on average in terms of AUC, outperforming previous state-of-the-art method GNNSafe.

**Observation 4: combining other GCL and MLE not works well** We try to replace the DGI algorithm with other widely used GCL algorithms: GRACE (Zhu et al., 2020b) and SUGRL (Mo et al., 2022) (without `MH`, `CE`, and `ERo`). From Tab. 5, DGI consistently outperforms its counterparties.

Table 5: The performance of different GCL algorithms, measured by the average AUC↑.

| | Cora | Twitch | Cham. | Cornell | Avg |
|---|---|---|---|---|---|
| GCN | 68.25 | 53.94 | 74.65 | 75.75 | 68.15 |
| GRACE | 83.01 | 65.78 | 87.99 | 80.76 | 79.38 |
| SUGRL | 82.41 | 72.15 | 82.71 | 63.67 | 75.24 |
| DGI | 88.87 | 84.77 | 90.63 | 87.00 | 87.82 |

From the above observation, we can conclude that the key reason behind the success of proposed DeGEM is due to **the symbiosis between DGI and EBM**. We extensively evaluate the potential variants, however all variants show significantly worse performance. This also aligns with our theoretical analysis in Sec. 3.5, which said that the learning of DGI can be viewed as learning EBM via variational learning. The property enables us to use DGI for learning the graph encoder as part of the final constructed EBMs.

## 5 RELATED WORK

**EBMs on Graph.** Some previous works (Hataya et al., 2021; Liu et al., 2021) apply EBMs on graphs that are trained with MLE have been proposed for graph generation. However, their works are performed on small graphs, where the over-computational issue of the sampling adjacency matrix is neglected, making them non-scalable. In contrast, our proposed CLEBM focuses on OOD detection, decomposing the EBM networks into informative representation extraction and output energy score based on given latent. The design enables us to move MCMC to latent space, which does not suffer from sampling adjacency matrix and therefore has excellent scalability.

**Graph OOD Detection.** OOD detection for non-graph data by neural networks has garnered considerable attention in the literature (Hendrycks & Gimpel, 2016; Hendrycks et al., 2019; Bevandić et al., 2018; Liu et al., 2020; Mohseni et al., 2020; Ren et al., 2019). However, these methods typically assume that instances (such as images) are i.i.d., overlooking scenarios with inter-dependent data that are common in many real-world applications. In contrast, Graph OOD Detection that inherently includes inter-dependent structures has not been explored well. Recently, some works (Li et al., 2022b; Bazhenov et al., 2022) focus on Graph OOD Detection on *graph-level*, i.e., detecting OOD graphs. These works treat each graph as an independent instance, while OOD detection on *node-level* presents unique challenges given the non-negligible inter-dependence between instances. To this end, Bayesian GNN models have been proposed that can detect OOD nodes within a graph by incorporating the inherent uncertainty in such inter-dependent data (Zhao et al., 2020a; Stadler et al., 2021a). OODGAT (Song & Wang, 2022) emphasizes the importance of node connection patterns for outlier detection, explicitly modeling node interactions and separating inliers from outliers during feature propagation. Energy-based Detection on graphs has been explored in GNNSafe (Wu et al., 2023b), by directly combing GNNs and Energy-based Detection on Images (Liu et al., 2020). However, their energy score is directly construed by classification, which is less effective. Moreover, they use energy propagation to enhance performance, which highly relies on the homophily assumption. Additionally, some methods train the model with *OOD Exposure*, i.e., training with both a known ID dataset and a known OOD dataset (Hendrycks et al., 2018; Liu et al., 2020; Wu et al., 2023b). In contrast, we decompose EBM learning into representation learning and energy learning, delivering better detection capability without OOD exposure (see Sec. 4.2). Additionally, benefits from powerful energy construction and effective graph encoder, we do not require energy propagation, thus keeping high performance across homophilic and heterophilic graphs.

## 6 CONCLUSION

We introduce a novel approach, **DeGEM**, for graph OOD node detection, overcoming the heterophily issue and the computational challenges associated with MCMC sampling in large graphs. By decoupling the learning process into a GNN-based graph encoder and an energy head, we managed to leverage the GCL algorithm and classification loss to learn robust node representations and perform efficient MCMC sampling in the latent space, circumventing the need to directly sample the adjacency matrix. The design of **DeGEM**, featuring a Multi-Hop Graph encoder and a Recurrent Update mechanism, facilitates the incorporation of topological information into node representations, which is crucial for OOD detection in graph-structured data. Extensive experimental evaluations have validated the effectiveness of **DeGEM**, which not only exhibits superior performance on both homophilic and heterophilic graphs compared to baselines with/without OOD exposure, but also outperforms methods trained with OOD exposure in a label-insufficient scenario.

## Acknowledgments

Xiaochun Cao's work is partially supported by the Shenzhen Science and Technology Program (No. KQTD20221101093559018) and by the National Natural Science Foundation of China (No. 62411540034). Pengwen Dai's work is supported by the National Natural Science Foundation of China (No. 62302532). Jing Tang's work is partially supported by National Key R&D Program of China under Grant No. 2023YFF0725100 and No. 2024YFA1012701, by the National Natural Science Foundation of China (NSFC) under Grant No. 62402410 and No. U22B2060, by Guangdong Provincial Project (No. 2023QN10X025), by Guangdong Basic and Applied Basic Research Foundation under Grant No. 2023A1515110131, by Guangzhou Municipal Science and Technology Bureau under Grant No. 2023A03J0667 and No. 2024A04J4454, by Guangzhou Municipal Education Bureau (No. 2024312263), and by Guangzhou Municipality Big Data Intelligence Key Lab (No. 2023A03J0012), Guangzhou Industrial Information and Intelligent Key Laboratory Project (No. 2024A03J0628) and Guangzhou Municipal Key Laboratory of Financial Technology Cutting-Edge Research (No. 2024A03J0630).

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

## A  POTENTIAL BROADER IMPACT

This paper presents work whose goal is to advance the field of Machine Learning. There are many potential societal consequences of our work, none of which we feel must be specifically highlighted here.

## B  LIMITATIONS

DeGEM has achieved excellent performance, but the results are nearly maxed out on some datasets. We need better and more challenging benchmarks to evaluate performance. Additionally, DeGEM shares the same drawback as EBMs trained with MLE: training requires MCMC sampling. However, it still shows superior performance. Besides, the cost of MCMC sampling can be reduced by cooperative learning (Xie et al., 2018; 2021; Luo et al., 2024b).

## C  ADDITIONAL RELATED WORKS

**Graph Contrastive Learning.** Contrastive learning (CL) stands as a widely applied self-supervised learning method, aiming to derive informative sample representation solely from feature information. The main idea of CL is to align the representations of similar samples in close proximity while driving apart the representations of dissimilar samples. Witnessing the remarkable advancement of Graph Neural Networks (GNNs), a substantial amount of recent research has focused on Graph Contrastive Learning (GCL) (Veličković et al., 2018; Peng et al., 2020; Hassani & Khasahmadi, 2020; Zhu et al., 2020b; Mo et al., 2022). DGI (Veličković et al., 2018) learns by maximizing mutual information between node representations and corresponding high-level summaries of graphs. GRACE (Zhu et al., 2020b) maximizes the agreement of corresponding node representations in two augmented views for a graph. SUGRL (Mo et al., 2022) explores the complementary information from structural and neighbor information to maximize inter-class variation and minimize intra-class variation through triplet losses and an upper bound loss, while removing the need for data augmentation and discriminators. Our work builds upon the foundation of GCL, where we employ DGI as part of learning the graph encoder that extract graph topology information. We also establish the relationship between DGI and EBM: DGI can be understood as an EBM trained by noise contrastive estimation. To further enhance the capabilities of the proposed method, we design recurrent update to let the energy head and DGI promote each other in training (see Sec. 3.3), and experiments show that this to be greatly beneficial for OOD detection (see Sec. 4.4).

**Node Anomaly Detection (NAD).** NAD is a binary classification task, that directly categorizes nodes into two different categories: normal and anomalous. Some works (Dong et al., 2025a; 2024; 2025b; Zhao & Akoglu, 2021; Gong et al., 2023) have been proposed to handle this problem. In contrast, node OOD detection requires balancing the ability to classify in-distribution nodes and detect out-of-distribution nodes.

## D  DERIVIATIONS

### D.1  TRAINING EBMS

Given a Boltzmann distribution $p_{\boldsymbol{\theta}}(\boldsymbol{x}) = \frac{\exp(-E_{\boldsymbol{\theta}}(\boldsymbol{x}))}{Z_{\boldsymbol{\theta}}}$, its negative log-likelihood (NLL) $\mathcal{L}_{\boldsymbol{\theta}}$ is:

$$\mathcal{L}_{\boldsymbol{\theta}} = -\mathbb{E}_{\boldsymbol{x} \sim p_d(\boldsymbol{x})}\Big[\log p_{\boldsymbol{\theta}}(\boldsymbol{x})\Big].$$

To minimize $\mathcal{L}_{\boldsymbol{\theta}}$, we first need to calculate the gradient of $\log p_{\boldsymbol{\theta}}(\boldsymbol{x})$ w.r.t. $\boldsymbol{\theta}$:

$$
\begin{aligned}
\nabla_{\boldsymbol{\theta}} \log p_{\boldsymbol{\theta}}(\boldsymbol{x}) &= \nabla_{\boldsymbol{\theta}} \log \frac{\exp\left(-E_{\boldsymbol{\theta}}(\boldsymbol{x})\right)}{Z_{\boldsymbol{\theta}}} \\
&= -\nabla_{\boldsymbol{\theta}} E_{\boldsymbol{\theta}}(\boldsymbol{x}) - \nabla_{\boldsymbol{\theta}} \log Z_{\boldsymbol{\theta}} \\
&= -\nabla_{\boldsymbol{\theta}} E_{\boldsymbol{\theta}}(\boldsymbol{x}) - \nabla_{\boldsymbol{\theta}} \log \sum_{\boldsymbol{x}} \exp\left(-E_{\boldsymbol{\theta}}(\boldsymbol{x})\right) \\
&= -\nabla_{\boldsymbol{\theta}} E_{\boldsymbol{\theta}}(\boldsymbol{x}) - \frac{1}{\sum_{\boldsymbol{x}'} \exp\left(-E_{\boldsymbol{\theta}}(\boldsymbol{x}')\right)} \sum_{\boldsymbol{x}} \nabla_{\boldsymbol{\theta}} \exp\left(-E_{\boldsymbol{\theta}}(\boldsymbol{x})\right) \\
&= -\nabla_{\boldsymbol{\theta}} E_{\boldsymbol{\theta}}(\boldsymbol{x}) + \sum_{\boldsymbol{x}} \frac{\exp\left(-E_{\boldsymbol{\theta}}(\boldsymbol{x})\right)}{\sum_{\boldsymbol{x}'} \exp\left(-E_{\boldsymbol{\theta}}(\boldsymbol{x}')\right)} \nabla_{\boldsymbol{\theta}} E_{\boldsymbol{\theta}}(\boldsymbol{x}) \\
&= -\nabla_{\boldsymbol{\theta}} E_{\boldsymbol{\theta}}(\boldsymbol{x}) + \sum_{\boldsymbol{x}} p_{\boldsymbol{\theta}}(\boldsymbol{x}) \nabla_{\boldsymbol{\theta}} E_{\boldsymbol{\theta}}(\boldsymbol{x}) \\
&= -\nabla_{\boldsymbol{\theta}} E_{\boldsymbol{\theta}}(\boldsymbol{x}) + \mathbb{E}_{\tilde{\boldsymbol{x}} \sim p_{\boldsymbol{\theta}}(\tilde{\boldsymbol{x}})}\left[\nabla_{\boldsymbol{\theta}} E_{\boldsymbol{\theta}}(\tilde{\boldsymbol{x}})\right].
\end{aligned}
$$

So, we have

$$
\nabla_{\boldsymbol{\theta}} \mathcal{L}_{\boldsymbol{\theta}} = -\nabla_{\boldsymbol{\theta}} \mathbb{E}_{\boldsymbol{x} \sim p_d(\boldsymbol{x})}\left[\log p_{\boldsymbol{\theta}}(\boldsymbol{x})\right] = \mathbb{E}_{\boldsymbol{x} \sim p_d(\boldsymbol{x})}\left[\nabla_{\boldsymbol{\theta}} E_{\boldsymbol{\theta}}(\boldsymbol{x})\right] - \mathbb{E}_{\tilde{\boldsymbol{x}} \sim p_{\boldsymbol{\theta}}(\tilde{\boldsymbol{x}})}\left[\nabla_{\boldsymbol{\theta}} E_{\boldsymbol{\theta}}(\tilde{\boldsymbol{x}})\right],
$$

Therefore, the EBM loss can be reformulated as:

$$
\mathcal{L}_{\boldsymbol{\theta}} = \mathbb{E}_{\boldsymbol{x} \sim p_d(\boldsymbol{x})}\left[E_{\boldsymbol{\theta}}(\boldsymbol{x})\right] - \mathbb{E}_{\tilde{\boldsymbol{x}} \sim p_{\boldsymbol{\theta}}(\tilde{\boldsymbol{x}})}\left[E_{\boldsymbol{\theta}}(\tilde{\boldsymbol{x}})\right] \approx \frac{1}{N} \sum_{i=1}^{N} E_{\boldsymbol{\theta}}(\boldsymbol{x}_i) - \frac{1}{M} \sum_{j=1}^{M} E_{\boldsymbol{\theta}}(\tilde{\boldsymbol{x}}_j).
$$

Additionally, minimizing $\mathcal{L}_{\boldsymbol{\theta}}$ is equivalent to minimizing the KL divergence $\mathrm{KL}\left(p_d(\boldsymbol{x}) \| p_{\boldsymbol{\theta}}(\boldsymbol{x})\right)$:

$$
\begin{aligned}
\nabla_{\boldsymbol{\theta}} \mathrm{KL}\left(p_d(\boldsymbol{x}) \| p_{\boldsymbol{\theta}}(\boldsymbol{x})\right) &= \nabla_{\boldsymbol{\theta}} \sum_{\boldsymbol{x}} p_d(\boldsymbol{x}) \log \frac{p_d(\boldsymbol{x})}{p_{\boldsymbol{\theta}}(\boldsymbol{x})} \\
&= \nabla_{\boldsymbol{\theta}} \sum_{\boldsymbol{x}} p_d(\boldsymbol{x}) \log p_d(\boldsymbol{x}) - \nabla_{\boldsymbol{\theta}} \sum_{\boldsymbol{x}} p_d(\boldsymbol{x}) \log p_{\boldsymbol{\theta}}(\boldsymbol{x}) \\
&= \nabla_{\boldsymbol{\theta}} \mathbb{E}_{\boldsymbol{x} \sim p_d(\boldsymbol{x})}\left[\log p_d(\boldsymbol{x})\right] - \nabla_{\boldsymbol{\theta}} \mathbb{E}_{\boldsymbol{x} \sim p_d(\boldsymbol{x})}\left[\log p_{\boldsymbol{\theta}}(\boldsymbol{x})\right] \\
&= \mathbf{0} + \nabla_{\boldsymbol{\theta}} \mathcal{L}_{\boldsymbol{\theta}} = \nabla_{\boldsymbol{\theta}} \mathcal{L}_{\boldsymbol{\theta}}.
\end{aligned}
$$

### D.2 REGULARIZATION FOR EBM LOSS

Additionally, we follow (Du & Mordatch, 2019) to adopt the $L_2$ regularization for energy magnitudes of both true data and sampled data when computing $\mathcal{L}_{\mathrm{ebm}}$ during training, as otherwise while the difference between true data and sampled data was preserved, the actual values would fluctuate to numerically unstable values. For an energy function $E_{\boldsymbol{\theta}}(\cdot)$, true data $\boldsymbol{x}_i \sim p_d(\boldsymbol{x})$, and sampled data $\tilde{\boldsymbol{x}}_j \sim p_{\boldsymbol{\theta}}(\tilde{\boldsymbol{x}})$:

$$
\mathcal{L}_{\mathrm{ebm}} = \frac{1}{N} \sum_{i=1}^{N} E_{\boldsymbol{\theta}}(\boldsymbol{x}_i) - \frac{1}{M} \sum_{j=1}^{M} E_{\boldsymbol{\theta}}(\tilde{\boldsymbol{x}}_j) + c \left[\frac{1}{N} \sum_{i=1}^{N} E_{\boldsymbol{\theta}}(\boldsymbol{x}_i)^2 + \frac{1}{M} \sum_{j=1}^{M} E_{\boldsymbol{\theta}}(\tilde{\boldsymbol{x}}_j)^2\right],
$$

where $c$ is a coefficient, we set $c = 1$ here.

### D.3 SAMPLING FROM EBM

Given a Boltzmann distribution $p_{\boldsymbol{\theta}}(\boldsymbol{x}) = \frac{\exp(-E_{\boldsymbol{\theta}}(\boldsymbol{x}))}{Z_{\boldsymbol{\theta}}}$, the gradient of $\log p_{\boldsymbol{\theta}}(\boldsymbol{x})$ w.r.t. $\boldsymbol{x}$ is:

$$
\begin{aligned}
\nabla_{\boldsymbol{x}} \log p_{\boldsymbol{\theta}}(\boldsymbol{x}) &= \nabla_{\boldsymbol{x}} \log \frac{\exp\left(-E_{\boldsymbol{\theta}}(\boldsymbol{x})\right)}{Z_{\boldsymbol{\theta}}} \\
&= -\nabla_{\boldsymbol{x}} E_{\boldsymbol{\theta}}(\boldsymbol{x}) - \nabla_{\boldsymbol{x}} \log Z_{\boldsymbol{\theta}} \\
&= -\nabla_{\boldsymbol{x}} E_{\boldsymbol{\theta}}(\boldsymbol{x}),
\end{aligned}
$$

---

**Algorithm 1** Training algorithm of DeGEM.

---

**Require:** graph $\mathcal{G} = (\mathbf{X}, \mathbf{A})$, learning rate $\eta$, classification loss weight $\xi$, MCMC step size $\lambda$,
  MCMC noise variance $\sigma^2$, MCMC steps $K$, MCMC samples $M$, epoch number $E$.
**Ensure:** optimized models $\{g_{\hat{\alpha}}, D_{\hat{\psi}}, f_{\hat{\omega}}, I_{\hat{\phi}}\}$.

1: Initialize weights $\{\boldsymbol{\alpha}_0, \boldsymbol{\psi}_0, \boldsymbol{\omega}_0, \boldsymbol{\phi}_0\}$;
2: Initialize MCMC replay buffer $\mathcal{B} \leftarrow \varnothing$;
3: **for** $e \leftarrow 1$ **to** $E$ **do**
4:   Get node embeddings $\{\boldsymbol{h}_i\}_{i=1}^N \leftarrow \{g_{\boldsymbol{\alpha}_{e-1}}(\boldsymbol{x}_i, \mathcal{N}(\boldsymbol{x}_i))\}_{i=1}^N$;
5:
6:   # MCMC sampling
7:   **for** sample node $j = 1$ **to** $M$ **do**
8:     $\tilde{\boldsymbol{h}}_j^{(0)} \sim \mathcal{B}$ with 95% probability and $\mathcal{U}$ otherwise;
9:     **for** sample step $k = 1$ **to** $K$ **do**
10:       $\tilde{\boldsymbol{h}}_j^{(k)} = \tilde{\boldsymbol{h}}_j^{(k-1)} - \lambda \nabla_{\tilde{\boldsymbol{h}}} f_{\boldsymbol{\omega}_{e-1}}(\tilde{\boldsymbol{h}}_j^{(k-1)}) + \boldsymbol{\epsilon}^{(k)}, \boldsymbol{\epsilon}^{(k)} \sim \mathcal{N}(0, \sigma^2)$;
11:     **end for**
12:     $\tilde{\boldsymbol{h}}_j \leftarrow \mathrm{sg}[\tilde{\boldsymbol{h}}_j^{(K)}]$;
13:     $\mathcal{B} \leftarrow \mathcal{B} \cup \{\tilde{\boldsymbol{h}}_j\}$;
14:   **end for**
15:   Sampled embeddings $\{\tilde{\boldsymbol{h}}_j\}_{j=1}^M$;
16:
17:   # compute losses
18:   Compute $\mathcal{L}_{\mathrm{ebm}}, \mathcal{L}_{\mathrm{cl}}$, and $\mathcal{L}_{\mathrm{cls}}$ via recurrent update (Sec. 3.4);
19:
20:   # update models
21:   $\boldsymbol{\alpha}_e \leftarrow \boldsymbol{\alpha}_{e-1} - \eta \nabla_{\boldsymbol{\alpha}}(\mathcal{L}_{\mathrm{cl}} + \xi \mathcal{L}_{\mathrm{cls}})$;
22:   $\boldsymbol{\psi}_e \leftarrow \boldsymbol{\psi}_{e-1} - \eta \nabla_{\boldsymbol{\psi}} \mathcal{L}_{\mathrm{cl}}$;
23:   $\boldsymbol{\omega}_e \leftarrow \boldsymbol{\omega}_{e-1} - \eta \nabla_{\boldsymbol{\omega}} \mathcal{L}_{\mathrm{ebm}}$;
24:   $\boldsymbol{\phi}_e \leftarrow \boldsymbol{\phi}_{e-1} - \eta \nabla_{\boldsymbol{\phi}}(\xi \mathcal{L}_{\mathrm{cls}})$;
25: **end for**
26: $\hat{\boldsymbol{\alpha}} \leftarrow \boldsymbol{\alpha}_E, \hat{\boldsymbol{\psi}} \leftarrow \boldsymbol{\psi}_E, \hat{\boldsymbol{\omega}} \leftarrow \boldsymbol{\omega}_E, \hat{\boldsymbol{\phi}} \leftarrow \boldsymbol{\phi}_E$.

---

so we can first initialize a sample using uniform distribution $\boldsymbol{x}^{(0)} \sim \mathcal{U}$, and utilize multi steps stochastic gradient to make the sample follow distribution $p_{\boldsymbol{\theta}}(\boldsymbol{x})$:

$$\boldsymbol{x}^{(k)} = \boldsymbol{x}^{(k-1)} + \lambda \nabla_{\boldsymbol{x}} \log p_{\boldsymbol{\theta}}(\boldsymbol{x}^{(k-1)}) + \boldsymbol{\epsilon}^{(k)}$$
$$= \boldsymbol{x}^{(k-1)} - \lambda \nabla_{\boldsymbol{x}} E_{\boldsymbol{\theta}}(\boldsymbol{x}^{(k-1)}) + \boldsymbol{\epsilon}^{(k)},$$

where $\boldsymbol{\epsilon}^{(k)} \sim \mathcal{N}(0, \sigma^2)$. After $K$ steps, $\tilde{x}^{(K)}$ is the output of $K$-step MCMC sampling.

## E  IMPLEMENTATION DETAILS

### E.1  TRAINING ALGORITHM

The detailed training algorithm is shown in Algorithm 1. We first obtain node embeddings (Line 4), and then sample nodes by a $K$-step MCMC sampling (Line 7-15). Then the recurrent update is conducted for computing losses $\mathcal{L}_{\mathrm{ebm}}, \mathcal{L}_{\mathrm{cl}}$, and $\mathcal{L}_{\mathrm{cls}}$ (Line 18). Finally, update the parameters (Line 21-24).

Furthermore, we follow (Du & Mordatch, 2019) to use sample **relay buffer**. The initialization of MCMC chain plays a crucial role in mixing time, but langevin dynamics does not place restrictions on sample initialization given sufficient sampling steps. Thus we use a sample replay buffer $\mathcal{B}$ in which we preserve previously generated samples and use either these samples or uniform noise for initialization.

Table 6: Search space of hyper-parameters.

|  | **Search Space** |
| --- | --- |
| learning rate $\eta$ | $\{0.05, 0.01, 0.001, 0.0005, 0.0001\}$ |
| weight decay | $\{0.0, 0.01, 0.001, 0.0005, 0.0001\}$ |
| dropout | $\{0.0, 0.1, 0.2, 0.3, 0.4, 0.5\}$ |
| $\beta$ | $\{0, 0.1, 0.3, 0.5, 0.7, 0.9, 1\}$ |
| $\rho$ | $\{0, 0.1, 0.3, 0.5, 0.7, 0.9, 1\}$ |
| $\gamma$ | $\{0, 0.1, 0.3, 0.5, 0.7, 0.9, 1\}$ |
| $\lambda$ | $\{1, 5, 10\}$ |
| $\sigma^2$ | $\{0.01, 0.005\}$ |
| $\xi$ | $\{0.01, 0.05, 0.1, 0.3, 0.5, 1.0\}$ |

### E.2 DATASET AND SPLITS

For `Twitch` that has several subgraphs, we use `DE` as ID dataset and the other five subgraphs as OOD datasets. The subgraph `ENGB` is used as OOD exposure dataset for training the OOD Expo baselines. These subgraphs are different in size, edge densities, and degree distribution, thus can be regarded as samples from different distribution (Wu et al., 2021).

For `Arxiv` that is in a single graph, we divide the nodes into three parts: the papers published before 2015 are used as ID data, those after 2017 are used as OOD data, and those between are used as OOD exposure data for training the OOD Expo baselines.

For `Cora`, `Amazon`, `Chameleon`, `Actor`, and `Cornell` that have no clear domain information, we synthesize OOD data in three ways (Wu et al., 2023a). Structure manipulation (S): use the original graph as ID data and adopt a stochastic block model to randomly generate a OOD graph. Feature interpolation (F): use random interpolation to create node features for OOD data and the original graph as ID data. Label leave-out (L): use nodes with partial classes as ID and leave out others for OOD.

We split the ID dataset as 10%/10%/80% (train/valid/test), and use all the nodes in OOD dataset for evaluation. Additionally, an extra OOD dataset is also used in training for OOD exposure (`OE`, `Energy-FT`, and `GNNSafe++`).

### E.3 SEARCH SPACE FOR HYPER-PARAMETERS

The detailed search space of hyper-parameters is shown in Tab. 6. We use Optuna (Akiba et al., 2019) to conduct random search for hyper-parameters.

## F COMPLEXITY ANALYSIS

Compared to other baselines, our additional computational complexity comes from the introduced GCL and MCMC sampling.

Suppose that the GCN is used as the graph encoder in other baselines, with $m$ the edge numbers, $n$ the node numbers, $d_0$ the initial node dimension, $d_1$ the latent dimension of node representation, and $L$ the number of layers (propagation numbers). So other baselines have a complexity of $\mathcal{O}(Lmd_0d_1)$.

In terms of our methods, the GCL processes both positive nodes and negative nodes, but the propagation for positive nodes is conducted in pre-process, with only the transformation remaining in training iterations. So the complexity for GCL is $\mathcal{O}(Lnd_0d_1 + Lmd_0d_1) \approx \mathcal{O}(Lmd_0d_1)$, since normally $m >> n$, where $n$ is the number of nodes. The MCMC sampling has a complexity of $\mathcal{O}((nd_1d_2 + nd_2)K) = \mathcal{O}(nd_1d_2K)$, where $d_2$ is the hidden dimension of Energy Function (2-layer MLP), and $K$ is the number of steps of MCMC sampling.

This actually suggests that our method has scalability comparable to GNNs, since the seemingly computationally intensive MCMC steps do not suffer from edges, with complexity only scaling linearly with the number of nodes.

Table 7: The average time consumption (s) of one-run DeGEM training, including the total time, the time consumption of GCL algorithm and MCMC sampling.

|  | Cora | Amazon | Twitch | Arxiv | Chameleon | Actor | Cornell | Avg | |
|---|---|---|---|---|---|---|---|---|---|
| Total | 11.64 | 15.26 | 15.54 | 149.17 | 10.40 | 17.71 | 5.52 | 32.18 | 100% |
| GCL | 0.71 | 0.90 | 0.67 | 10.08 | 0.58 | 1.02 | 0.18 | 2.02 | 6% |
| MCMC | 5.15 | 5.45 | 4.18 | 15.12 | 4.53 | 6.82 | 2.61 | 6.27 | 19% |

Table 8: The average training time consumption (s) (for one-run) comparison between baselines and DeGEM.

|  | Cora | Amazon | Twitch | Arxiv | Chameleon | Actor | Cornell | Avg |
|---|---|---|---|---|---|---|---|---|
| MSP | 3.95 | 17.98 | 24.45 | 104.16 | 5.87 | 6.35 | 3.16 | 23.70 |
| ODIN | 6.43 | 31.65 | 42.72 | 225.78 | 10.23 | 10.68 | 5.38 | 47.55 |
| Mahalanobis | 28.77 | 96.66 | 187.42 | 4956.43 | 36.32 | 71.06 | 20.40 | 771.01 |
| Energy | 4.81 | 21.40 | 26.36 | 117.02 | 7.14 | 7.78 | 4.10 | 26.94 |
| GKDE | 5.21 | 10.58 | 14.15 | 45.06 | 5.37 | 7.72 | 4.48 | 13.22 |
| GPN | 15.28 | 27.88 | 23.55 | OOM | 14.39 | 22.51 | 11.43 | N/A |
| OODGAT | 5.18 | 23.13 | 33.80 | 144.25 | 7.78 | 8.70 | 4.56 | 32.49 |
| GNNSafe | 5.59 | 22.28 | 26.42 | 136.55 | 7.55 | 8.12 | 3.83 | 30.05 |
| OE | 5.72 | 24.40 | 27.39 | 129.01 | 7.63 | 8.38 | 4.27 | 29.54 |
| Energy-FT | 6.34 | 25.06 | 28.37 | 131.99 | 8.44 | 9.30 | 5.14 | 30.66 |
| GNNSafe++ | 8.23 | 25.68 | 31.14 | 138.09 | 9.66 | 10.43 | 7.13 | 32.91 |
| DeGEM | 11.64 | 15.26 | 15.54 | 149.17 | 10.40 | 17.71 | 5.52 | 32.18 |

Table 9: Average OOD detection performance measured by AUC↑ on synthetic-Cora datasets with various homophily level.

| homo ratio | 0.0 | 0.2 | 0.4 | 0.6 | 0.8 | 1.0 |
|---|---|---|---|---|---|---|
| GNNSafe | 69.02 | 70.65 | 72.73 | 73.72 | 79.08 | 79.12 |
| DeGEM | 99.66 | 96.56 | 98.39 | 99.11 | 99.86 | 99.96 |

# G  TIME CONSUMPTION

The detailed time consumption of DeGEM training is shown in Tab. 7. In the GCL algorithm, message-passing for positive samples is handled during preprocessing, and the augmentation step only involves shuffling the rows of the node feature matrix. This makes the entire GCL learning process highly efficient. Furthermore, MCMC sampling is conducted in the latent space, which is low-dimensional and topology-free, resulting in an acceptable time consumption. Overall, GCL learning and MCMC sampling account for only 6% and 19% of the total training time, respectively.

Tab. 8 compares the training time consumption (s) of baseline models and our proposed DeGEM. Thanks to preprocessing node features and MCMC sampling in the latent space, DeGEM requires only slightly more time during the training phase compared to other baselines. Specifically, training DeGEM (32.18s) is just 7% slower than GNNSafe (30.66s), while delivering significant performance improvements, particularly on heterophilic graphs.

# H  EVALUATION ON SYNTHETIC GRAPHS

We conduct additional experiments on varying the homophilic ratio using synthetic graphs (Zhu et al., 2021). We report the AUROC results on synthetic-Cora in Tab. 9, and we use the Feature OOD setting.

As the homophilic ratio decreases from 1.0 to 0.0, the performance of GNNSafe decreases significantly from 79.12 to 69.02. In contrast, DeGEM consistently maintains a high performance across homophilic ratios, achieving around 99 in most cases. This indicates that GNNSafe struggles to handle heterophily, whereas our proposed method demonstrates strong performance on both homophilic and heterophilic graphs.

Table 10: Detailed OOD detection performance measured by AUC↑ / AUPR↑ / FPR95↓ / Acc↑ on **homophilic** datasets with three OOD types (Structure manipulation, Feature interpolation, Label leave-out). The cells filled in gray are methods tailored for graph-format inputs.

| Methods | OOD Expo | Structure | | | | Feature | | | | Label | | | | Avg | | | |
|---|---|---|---|---|---|---|---|---|---|---|---|---|---|---|---|---|---|
| | | AUC↑ | AUPR↑ | FPR95↓ | Acc↑ | AUC↑ | AUPR↑ | FPR95↓ | Acc↑ | AUC↑ | AUPR↑ | FPR95↓ | Acc↑ | AUC↑ | AUPR↑ | FPR95↓ | Acc↑ |
| **Cora** | | | | | | | | | | | | | | | | | |
| MSP | No | 75.05 | 52.23 | 85.23 | 77.30 | 85.87 | 75.26 | 65.29 | 77.30 | 91.10 | 78.24 | 43.41 | 88.29 | 84.01 | 68.57 | 64.64 | 80.96 |
| ODIN | No | 30.57 | 19.28 | 97.90 | 74.60 | 21.19 | 17.02 | 99.56 | 77.50 | 20.29 | 15.11 | 99.90 | 87.66 | 24.02 | 17.14 | 99.12 | 79.92 |
| Mahalanobis | No | 41.03 | 23.12 | 99.41 | 71.90 | 63.92 | 41.65 | 92.43 | 74.20 | 67.45 | 39.82 | 87.53 | 88.92 | 57.47 | 34.86 | 93.12 | 78.34 |
| Energy | No | 79.48 | 54.22 | 66.36 | 79.10 | 89.34 | 79.96 | 55.98 | 79.30 | 93.26 | 82.42 | 33.67 | 90.19 | 87.36 | 72.20 | 52.00 | 82.86 |
| ResidualFlow | No | 61.23 | 40.72 | 92.21 | 82.20 | 60.72 | 42.25 | 94.68 | 82.20 | 64.69 | 66.52 | 90.77 | 82.20 | 62.22 | 49.83 | 92.55 | 82.20 |
| GKDE | No | 84.49 | 60.76 | 56.57 | 78.00 | 90.75 | 82.21 | 44.20 | 81.70 | 94.59 | 84.66 | 21.10 | 91.77 | 89.94 | 75.88 | 40.62 | 83.82 |
| GPN | No | 82.21 | 57.01 | 56.50 | 81.00 | 88.06 | 73.28 | 43.61 | 78.80 | 91.74 | 85.71 | 51.62 | 91.77 | 87.34 | 72.00 | 50.58 | 83.86 |
| OODGAT | No | 53.75 | 30.58 | 96.38 | 34.90 | 57.03 | 37.64 | 98.89 | 14.50 | 95.57 | 88.58 | 19.68 | 89.56 | 68.78 | 52.27 | 71.65 | 46.32 |
| GNNSafe | No | 87.98 | 79.29 | 76.48 | 75.30 | 92.18 | 86.48 | 49.41 | 75.40 | 92.36 | 81.55 | 34.48 | 88.92 | 90.84 | 82.44 | 53.46 | 79.87 |
| OE | Yes | 74.42 | 53.05 | 83.20 | 71.60 | 86.58 | 77.23 | 66.03 | 74.10 | 89.06 | 76.45 | 52.33 | 88.61 | 83.35 | 68.91 | 67.19 | 78.10 |
| Energy-FT | Yes | 82.34 | 57.36 | 62.67 | 79.20 | 91.13 | 81.01 | 40.10 | 78.60 | 93.68 | 85.94 | 31.44 | 91.14 | 89.05 | 74.77 | 44.74 | 82.98 |
| GNNSafe++ | Yes | 91.11 | 82.69 | 53.58 | 76.90 | 94.51 | 89.03 | 32.90 | 76.50 | 92.45 | 82.13 | 35.40 | 90.51 | 92.69 | 84.62 | 40.63 | 81.30 |
| **DeGEM (Ours)** | No | **99.93** | **99.81** | **0.33** | **84.20** | **99.84** | **99.50** | **0.66** | **84.30** | **97.58** | **93.61** | **12.27** | **93.04** | **99.12** | **97.64** | **4.42** | **87.18** |
| **Amazon-Photo** | | | | | | | | | | | | | | | | | |
| MSP | No | 98.94 | 99.07 | 2.46 | 93.64 | 97.02 | 95.39 | 10.93 | 92.89 | 95.49 | 94.13 | 27.77 | 95.84 | 97.15 | 96.20 | 13.72 | 94.13 |
| ODIN | No | 3.50 | 26.70 | 100.00 | 91.42 | 5.31 | 26.92 | 99.74 | 92.91 | 10.16 | 24.55 | 99.89 | 96.08 | 6.33 | 26.06 | 99.88 | 93.47 |
| Mahalanobis | No | 62.40 | 70.20 | 99.75 | 93.42 | 72.47 | 66.92 | 83.93 | 92.88 | 60.80 | 47.36 | 79.17 | 95.84 | 65.22 | 61.49 | 87.62 | 94.04 |
| Energy | No | 99.94 | 99.92 | 0.07 | 93.04 | 98.51 | 96.39 | 3.83 | 92.76 | 97.13 | 96.11 | 14.35 | 95.68 | 98.53 | 97.47 | 6.08 | 93.83 |
| ResidualFlow | No | 82.58 | 85.75 | 94.51 | 93.55 | 75.46 | 80.42 | 99.45 | 93.55 | 78.31 | 89.08 | 98.20 | 93.55 | 78.78 | 85.08 | 97.39 | 93.55 |
| GKDE | No | 92.68 | 94.80 | 74.22 | 89.69 | 54.52 | 49.48 | 94.00 | 31.90 | 76.35 | 75.89 | 87.07 | 85.55 | 74.52 | 73.39 | 85.10 | 69.04 |
| GPN | No | 90.35 | 90.53 | 57.53 | 82.63 | 86.47 | 85.47 | 82.60 | 63.38 | 89.90 | 84.64 | 37.93 | 89.30 | 88.90 | 86.88 | 59.35 | 78.44 |
| OODGAT | No | 71.41 | 71.93 | 100.00 | 25.42 | 70.93 | 74.80 | 98.54 | 25.21 | 99.18 | 98.42 | 2.34 | 96.16 | 80.51 | 81.72 | 66.96 | 48.93 |
| GNNSafe | No | 98.69 | 99.26 | 0.00 | 93.74 | 98.47 | 98.91 | 0.39 | 92.96 | 97.34 | 96.96 | 3.89 | 95.72 | 98.16 | 98.38 | 1.43 | 94.14 |
| OE | Yes | 99.84 | 99.82 | 0.09 | 92.89 | 98.60 | 96.51 | 3.46 | 92.84 | 96.58 | 95.39 | 16.63 | 95.96 | 98.34 | 97.24 | 6.73 | 93.90 |
| Energy-FT | Yes | 99.97 | 99.97 | 0.00 | 93.63 | 98.88 | 96.98 | 2.72 | 90.26 | 98.37 | 97.23 | 6.56 | 87.44 | 99.07 | 98.06 | 3.09 | 90.44 |
| GNNSafe++ | Yes | 99.60 | 99.77 | 0.00 | 93.04 | 99.57 | 99.42 | 0.17 | 92.76 | 97.57 | 97.58 | 5.36 | 95.52 | 98.91 | 98.92 | 1.84 | 93.77 |
| **DeGEM (Ours)** | No | **100.00** | **100.00** | **0.00** | **94.49** | **99.91** | **99.86** | **0.41** | **93.97** | **99.28** | **98.62** | **1.82** | **95.80** | **99.73** | **99.49** | **0.74** | **94.76** |

| Methods | OOD Expo | ES | | | FR | | | RU | | | Avg | | | Acc↑ |
|---|---|---|---|---|---|---|---|---|---|---|---|---|---|---|
| | | AUC↑ | AUPR↑ | FPR95↓ | AUC↑ | AUPR↑ | FPR95↓ | AUC↑ | AUPR↑ | FPR95↓ | AUC↑ | AUPR↑ | FPR95↓ | |
| **Twitch** | | | | | | | | | | | | | | |
| MSP | No | 66.08 | 76.56 | 91.50 | 47.40 | 53.65 | 95.85 | 57.41 | 71.21 | 93.75 | 56.97 | 67.14 | 93.70 | 63.89 |
| ODIN | No | 43.97 | 58.89 | 97.48 | 51.84 | 55.13 | 96.15 | 49.53 | 63.70 | 96.99 | 48.45 | 59.24 | 96.88 | 62.05 |
| Mahalanobis | No | 46.73 | 58.91 | 93.98 | 49.69 | 53.28 | 95.79 | 38.38 | 56.34 | 97.83 | 44.94 | 56.18 | 95.87 | 62.95 |
| Energy | No | 58.42 | 68.55 | 90.45 | 72.91 | 75.26 | 80.48 | 69.90 | 79.26 | 86.11 | 67.07 | 74.36 | 85.68 | 65.59 |
| ResidualFlow | No | 62.60 | 74.31 | 93.20 | 55.59 | 61.24 | 95.15 | 67.68 | 77.84 | 88.14 | 61.96 | 71.13 | 92.16 | 66.37 |
| GKDE | No | 57.52 | 67.30 | 91.93 | 57.48 | 61.28 | 93.97 | 46.25 | 61.51 | 96.28 | 53.75 | 63.36 | 94.06 | 60.42 |
| GPN | No | 84.07 | 76.46 | 83.99 | 76.32 | 84.37 | 93.97 | 78.36 | 88.86 | 89.33 | 79.59 | 88.38 | 89.10 | 60.29 |
| OODGAT | No | 77.35 | 84.45 | 82.44 | 77.72 | 80.06 | 78.83 | 73.24 | 81.12 | 83.26 | 76.10 | 81.88 | 81.51 | 60.29 |
| GNNSafe | No | 51.00 | 57.41 | 80.79 | 79.08 | 81.54 | 68.51 | 82.93 | 86.45 | 57.08 | 71.00 | 75.13 | 68.79 | 66.18 |
| OE | Yes | 64.52 | 77.11 | 88.47 | 52.99 | 62.00 | 92.81 | 69.64 | 81.48 | 84.01 | 62.38 | 73.53 | 88.43 | 64.71 |
| Energy-FT | Yes | 93.55 | 96.62 | 42.86 | 70.89 | 75.99 | 87.47 | 97.71 | 98.69 | 10.58 | 87.38 | 90.43 | 46.97 | 63.82 |
| GNNSafe++ | Yes | 97.75 | 98.93 | 9.70 | 92.11 | 95.22 | 64.57 | 99.87 | 99.94 | 0.00 | 96.58 | 98.03 | 24.76 | 65.83 |
| **DeGEM (Ours)** | No | 94.83 | 97.63 | 43.80 | **97.36** | **98.58** | **2.53** | 94.76 | 97.57 | 41.00 | 95.65 | 97.93 | 29.11 | 64.51 |
| **Arxiv** | | | | | | | | | | | | | | |
| | | 2018 | | | 2019 | | | 2020 | | | Avg | | | Acc↑ |
| | | AUC↑ | AUPR↑ | FPR95↓ | AUC↑ | AUPR↑ | FPR95↓ | AUC↑ | AUPR↑ | FPR95↓ | AUC↑ | AUPR↑ | FPR95↓ | |
| MSP | No | 61.76 | 70.21 | 89.09 | 63.52 | 65.89 | 87.96 | 67.06 | 90.63 | 85.74 | 64.11 | 75.58 | 87.60 | 53.87 |
| ODIN | No | 44.01 | 55.17 | 97.85 | 42.63 | 47.18 | 98.28 | 38.96 | 78.45 | 99.37 | 41.87 | 60.27 | 98.50 | 49.29 |
| Mahalanobis | No | 57.08 | 65.09 | 93.69 | 56.76 | 57.85 | 94.01 | 56.92 | 85.95 | 95.01 | 56.92 | 69.63 | 94.24 | 51.59 |
| Energy | No | 64.61 | 72.30 | 88.40 | 65.90 | 67.30 | 87.44 | 70.37 | 91.85 | 85.35 | 66.96 | 77.82 | 87.06 | 53.92 |
| ResidualFlow | No | 62.38 | 71.02 | 91.09 | 63.82 | 67.23 | 91.26 | 62.21 | 89.22 | 92.74 | 62.81 | 75.83 | 91.70 | 54.84 |
| GKDE | No | 69.44 | 76.46 | 86.37 | 71.32 | 73.79 | 86.82 | 71.73 | 92.23 | 86.50 | 70.83 | 80.83 | 86.56 | 21.15 |
| GPN | No | OOM | OOM | OOM | OOM | OOM | OOM | OOM | OOM | OOM | OOM | OOM | OOM | OOM |
| OODGAT | No | 72.35 | 75.38 | 77.05 | 73.97 | 72.32 | 77.89 | 72.30 | 91.22 | 79.16 | 72.87 | 79.64 | 78.03 | 54.69 |
| GNNSafe | No | 67.27 | 75.46 | 87.17 | 69.20 | 72.41 | 85.99 | 79.02 | 94.91 | 80.69 | 71.83 | 80.93 | 84.62 | 54.26 |
| OE | Yes | 70.51 | 77.38 | 81.45 | 71.75 | 73.47 | 80.19 | 75.12 | 93.24 | 77.06 | 72.46 | 81.37 | 79.56 | 53.17 |
| Energy-FT | Yes | 78.93 | 83.99 | 68.81 | 79.13 | 80.19 | 68.71 | 82.08 | 95.44 | 66.21 | 80.04 | 86.54 | 67.91 | 39.26 |
| GNNSafe++ | Yes | 75.76 | 82.15 | 75.34 | 76.95 | 79.18 | 73.55 | 83.82 | 96.05 | 66.18 | 78.84 | 85.79 | 71.69 | 48.05 |
| **DeGEM (Ours)** | No | **81.30** | 81.20 | **42.03** | **86.00** | **80.50** | **31.41** | **86.01** | 94.71 | **31.28** | **84.44** | **85.47** | **34.91** | **58.20** |

# I DETAILED EVALUATION RESULTS

The detailed results of OOD detection on homophilic and heterophilic graphs are shown in Tabs. 10 and 11, respectively.

The detailed results of OOD detection on varying the available label ratio are shown in Tab. 12.

The full results of the ablation study regarding the different components are shown in Tab. 13.

The detailed results of OOD detection performance of different GCL algorithms are shown in Tab. 14.

Table 11: Detailed OOD detection performance measured by AUC↑ / AUPR↑ / FPR95↓ / Acc↑ on **heterophilic** datasets with three OOD types (Structure manipulation, Feature interpolation, Label leave-out). The cells filled in gray are methods tailored for graph-format inputs.

| Methods | OOD Expo | Structure | | | | Feature | | | | Label | | | | Avg | | | |
|---|---|---|---|---|---|---|---|---|---|---|---|---|---|---|---|---|---|
| | | AUC↑ | AUPR↑ | FPR95↓ | Acc↑ | AUC↑ | AUPR↑ | FPR95↓ | Acc↑ | AUC↑ | AUPR↑ | FPR95↓ | Acc↑ | AUC↑ | AUPR↑ | FPR95↓ | Acc↑ |
| **Chameleon** | | | | | | | | | | | | | | | | | |
| MSP | No | 99.28 | 99.33 | 1.41 | 33.41 | 70.92 | 71.75 | 86.39 | 32.42 | 53.32 | 71.78 | 91.67 | 43.89 | 74.51 | 80.95 | 59.82 | 36.57 |
| ODIN | No | 60.09 | 70.22 | 100.00 | 28.80 | 53.20 | 49.82 | 91.13 | 33.24 | 68.07 | 81.48 | 74.78 | 37.01 | 60.45 | 67.17 | 88.64 | 33.02 |
| Mahalanobis | No | 99.59 | 99.57 | 0.92 | 29.90 | 58.93 | 55.55 | 97.80 | 33.41 | 42.42 | 67.25 | 82.46 | 44.54 | 66.98 | 74.13 | 60.39 | 35.95 |
| Energy | No | 91.94 | 95.28 | 100.00 | 37.14 | 67.75 | 66.73 | 84.15 | 37.58 | 59.92 | 75.25 | 70.18 | 41.69 | 73.20 | 79.09 | 84.77 | 38.80 |
| ResidualFlow | No | 48.24 | 42.46 | 99.39 | 34.50 | 53.82 | 53.61 | 99.91 | 34.50 | 56.16 | 84.94 | 99.34 | 34.50 | 52.74 | 60.34 | 99.55 | 34.50 |
| GKDE | No | 96.06 | 96.66 | 18.45 | 30.12 | 67.33 | 70.27 | 91.26 | 34.94 | 60.22 | 77.42 | 74.78 | 40.40 | 74.54 | 81.45 | 61.50 | 35.15 |
| GPN | No | 82.90 | 87.44 | 100.00 | 20.41 | 64.99 | 57.79 | 88.76 | 30.99 | 72.68 | 82.83 | 77.85 | 34.71 | 73.52 | 76.02 | 88.87 | 28.70 |
| OODGAT | No | 54.89 | 52.07 | 97.45 | 26.06 | 53.86 | 49.47 | 97.28 | 29.73 | 65.33 | 74.23 | 78.29 | 40.68 | 58.03 | 58.59 | 91.01 | 32.16 |
| GNNSafe | No | 34.36 | 56.86 | 100.00 | 35.33 | 57.46 | 58.24 | 95.83 | 38.07 | 52.18 | 73.72 | 85.09 | 43.43 | 48.00 | 62.94 | 93.64 | 38.94 |
| OE | Yes | 98.67 | 98.83 | 3.60 | 34.89 | 69.00 | 65.65 | 82.26 | 35.49 | 75.54 | 84.02 | 70.83 | 40.22 | 81.07 | 82.83 | 52.23 | 36.87 |
| Energy-FT | Yes | 98.95 | 99.15 | 0.61 | 38.01 | 65.19 | 64.56 | 94.51 | 38.89 | 75.53 | 84.45 | 66.89 | 39.30 | 79.89 | 82.72 | 54.00 | 38.74 |
| GNNSafe++ | Yes | 36.23 | 58.23 | 100.00 | 36.59 | 57.80 | 58.51 | 93.94 | 37.41 | 50.65 | 73.38 | 87.06 | 42.33 | 48.23 | 63.37 | 93.67 | 38.78 |
| **DeGEM (Ours)** | No | **99.99** | **99.98** | **0.00** | **57.82** | **99.70** | **99.51** | **1.10** | **57.93** | **89.68** | **94.62** | **39.04** | **64.46** | **96.46** | **98.04** | **13.38** | **60.07** |
| **Actor** | | | | | | | | | | | | | | | | | |
| MSP | No | 71.38 | 72.10 | 92.30 | 21.64 | 59.11 | 53.90 | 92.04 | 24.34 | 56.32 | 85.03 | 89.21 | 37.27 | 62.27 | 70.35 | 91.19 | 27.75 |
| ODIN | No | 42.32 | 43.97 | 99.66 | 22.30 | 52.57 | 47.79 | 93.55 | 23.90 | 65.10 | 90.12 | 85.58 | 35.79 | 53.33 | 60.62 | 92.93 | 27.33 |
| Mahalanobis | No | 79.96 | 74.27 | 70.51 | 24.23 | 65.94 | 58.33 | 85.08 | 23.72 | 52.58 | 82.78 | 87.92 | 31.98 | 66.16 | 71.79 | 81.17 | 26.64 |
| Energy | No | 64.26 | 64.61 | 98.16 | 25.21 | 51.71 | 48.17 | 96.68 | 24.33 | 55.00 | 84.54 | 89.80 | 36.71 | 56.99 | 65.77 | 94.88 | 28.75 |
| ResidualFlow | No | 49.72 | 45.67 | 97.96 | 27.63 | 50.07 | 45.13 | 97.47 | 27.63 | 50.68 | 87.99 | 97.30 | 27.63 | 50.15 | 59.60 | 97.58 | 27.63 |
| GKDE | No | 71.27 | 69.91 | 93.39 | 25.63 | 58.08 | 52.60 | 90.67 | 19.56 | 53.72 | 84.44 | 92.50 | 33.60 | 61.02 | 68.98 | 92.19 | 26.26 |
| GPN | No | 78.58 | 79.13 | 62.50 | 18.67 | 62.13 | 56.27 | 93.42 | 20.21 | 75.04 | 93.13 | 75.73 | 38.08 | 71.92 | 76.18 | 77.22 | 25.65 |
| OODGAT | No | 51.25 | 46.46 | 94.83 | 23.65 | 52.00 | 48.78 | 95.20 | 25.77 | 65.39 | 88.68 | 84.64 | 36.30 | 56.21 | 61.31 | 91.56 | 28.57 |
| GNNSafe | No | 31.76 | 42.55 | 99.05 | 26.30 | 50.66 | 46.38 | 95.62 | 26.20 | 51.60 | 83.80 | 87.81 | 37.92 | 44.68 | 57.58 | 94.16 | 30.14 |
| OE | Yes | 68.52 | 67.55 | 94.59 | 21.61 | 61.52 | 56.27 | 94.84 | 24.64 | 60.54 | 86.41 | 83.00 | 37.11 | 63.53 | 70.07 | 90.81 | 27.79 |
| Energy-FT | Yes | 74.36 | 74.16 | 89.74 | 24.67 | 61.43 | 54.84 | 90.87 | 25.39 | 58.75 | 85.85 | 88.04 | 37.34 | 64.85 | 71.62 | 89.55 | 29.13 |
| GNNSafe++ | Yes | 58.49 | 63.44 | 98.89 | 25.13 | 62.64 | 62.03 | 93.49 | 25.31 | 52.36 | 84.47 | 89.92 | 37.22 | 57.83 | 69.98 | 94.10 | 29.22 |
| **DeGEM (Ours)** | No | **99.76** | **99.65** | **0.84** | **31.97** | **99.98** | **99.98** | **0.08** | **33.87** | **100.00** | **100.00** | **0.00** | 36.02 | **99.92** | **99.88** | **0.31** | **33.95** |
| **Cornell** | | | | | | | | | | | | | | | | | |
| MSP | No | 81.50 | 84.58 | 92.35 | 41.50 | 71.89 | 61.46 | 84.15 | 43.54 | 68.50 | 85.15 | 89.47 | 63.81 | 111.98 | 77.06 | 88.66 | 49.61 |
| ODIN | No | 67.99 | 57.50 | 98.91 | 38.78 | 85.76 | 87.82 | 75.96 | 38.10 | 42.16 | 66.71 | 89.47 | 63.81 | 94.76 | 70.68 | 88.11 | 46.89 |
| Mahalanobis | No | 69.84 | 69.08 | 92.90 | 36.73 | 70.08 | 69.09 | 92.90 | 43.54 | 81.73 | 93.72 | 97.37 | 62.86 | 104.50 | 77.30 | 94.39 | 47.71 |
| Energy | No | 83.09 | 84.90 | 86.34 | 38.10 | 86.70 | 85.61 | 83.06 | 38.10 | 69.70 | 85.99 | 100.00 | 62.86 | 121.88 | 85.50 | 89.80 | 46.35 |
| ResidualFlow | No | 67.80 | 64.21 | 93.44 | 43.54 | 66.33 | 61.30 | 98.36 | 43.54 | 73.09 | 91.56 | 97.37 | 43.54 | 88.58 | 72.35 | 96.39 | 43.54 |
| GKDE | No | 80.53 | 80.81 | 91.26 | 14.97 | 77.48 | 72.00 | 98.36 | 42.86 | 81.18 | 92.76 | 86.84 | 63.81 | 109.97 | 81.86 | 92.15 | 40.54 |
| GPN | No | 89.68 | 90.86 | 61.20 | 43.54 | 83.36 | 75.12 | 37.70 | 42.86 | 82.93 | 92.79 | 60.53 | 63.81 | 168.50 | 86.26 | 53.14 | 50.07 |
| OODGAT | No | 67.16 | 66.13 | 97.81 | 42.18 | 69.42 | 61.27 | 99.45 | 42.86 | 68.52 | 82.45 | 81.58 | 64.76 | 95.30 | 69.95 | 92.95 | 49.93 |
| GNNSafe | No | 74.66 | 82.14 | 93.44 | 25.17 | 76.22 | 83.44 | 88.52 | 41.50 | 68.17 | 80.87 | 68.42 | 63.81 | 115.19 | 82.15 | 83.46 | 43.49 |
| OE | Yes | 82.05 | 79.76 | 74.32 | 26.53 | 77.29 | 62.74 | 55.74 | 37.41 | 55.79 | 74.93 | 92.11 | 57.14 | 110.50 | 72.47 | 74.05 | 40.36 |
| Energy-FT | Yes | 80.42 | 83.71 | 92.90 | 43.54 | 90.19 | 90.55 | 71.58 | 39.46 | 65.81 | 83.53 | 89.47 | 62.86 | 128.71 | 85.93 | 84.65 | 48.62 |
| GNNSafe++ | Yes | 77.59 | 83.44 | 89.07 | 26.53 | 82.94 | 87.28 | 98.36 | 43.54 | 76.77 | 84.57 | 60.53 | 60.95 | 125.22 | 85.10 | 82.65 | 43.67 |
| **DeGEM (Ours)** | No | **97.97** | **96.11** | **5.46** | **48.30** | **100.00** | **100.00** | **0.00** | **65.31** | **90.90** | **96.74** | 65.79 | **77.14** | **233.73** | **97.61** | **23.75** | **63.58** |

Table 12: Detailed OOD detection performance on varying the available label ratio, measured by AUC↑ / AUPR↑ / FPR95↓ / Acc↑.

**Cora**

| | OOD Expo | Structure | | | | Feature | | | | Label | | | | Avg | | | |
|---|---|---|---|---|---|---|---|---|---|---|---|---|---|---|---|---|---|
| | | AUC↑ | AUPR↑ | FPR95↓ | Acc↑ | AUC↑ | AUPR↑ | FPR95↓ | Acc↑ | AUC↑ | AUPR↑ | FPR95↓ | Acc↑ | AUC↑ | AUPR↑ | FPR95↓ | Acc↑ |
| Label Rate = 10% | | | | | | | | | | | | | | | | | |
| OE | Yes | 51.47 | 30.28 | 93.09 | 42.50 | 64.60 | 42.82 | 89.11 | 49.30 | 80.01 | 67.94 | 83.87 | 74.37 | 65.36 | 47.01 | 88.69 | 55.39 |
| Energy-FT | Yes | 63.03 | 37.80 | 92.84 | 51.90 | 65.89 | 47.46 | 96.60 | 54.90 | 87.08 | 73.80 | 62.37 | 68.99 | 72.00 | 53.02 | 83.94 | 58.60 |
| GNNSafe++ | Yes | 76.10 | 56.67 | 81.72 | 52.50 | 72.84 | 60.39 | 97.75 | 52.90 | 87.27 | 76.58 | 58.72 | 77.85 | 78.74 | 64.55 | 79.40 | 61.08 |
| **DeGEM** | No | **99.50** | **98.25** | **1.92** | **61.20** | **98.78** | **96.23** | **5.17** | **60.80** | **95.35** | **89.90** | **22.92** | **80.70** | **97.87** | **94.79** | **10.00** | **67.57** |
| Label Rate = 50% | | | | | | | | | | | | | | | | | |
| OE | Yes | 67.64 | 45.88 | 86.96 | 68.30 | 82.14 | 67.48 | 66.06 | 71.60 | 86.12 | 68.08 | 59.74 | 87.66 | 78.63 | 60.48 | 70.92 | 75.85 |
| Energy-FT | Yes | 67.91 | 42.20 | 79.99 | 70.00 | 77.13 | 59.17 | 71.16 | 70.80 | 87.69 | 68.26 | 45.54 | 89.24 | 77.58 | 56.54 | 65.56 | 76.68 |
| GNNSafe++ | Yes | 82.22 | 67.91 | 74.19 | 70.60 | 87.76 | 76.69 | 60.97 | 71.90 | 89.43 | 73.81 | 37.02 | 88.61 | 86.47 | 72.80 | 57.39 | 77.04 |
| **DeGEM** | No | **98.46** | **95.88** | **6.76** | **79.90** | **98.72** | **97.08** | **4.95** | **78.40** | **95.40** | **82.90** | **14.91** | **91.14** | **97.53** | **91.95** | **8.87** | **83.15** |
| Label Rate = 100% | | | | | | | | | | | | | | | | | |
| OE | Yes | 74.42 | 53.05 | 83.20 | 71.60 | 86.58 | 77.23 | 66.03 | 74.10 | 89.06 | 76.45 | 52.33 | 88.61 | 83.35 | 68.91 | 67.19 | 78.10 |
| Energy-FT | Yes | 82.34 | 57.36 | 62.67 | 79.20 | 91.13 | 81.01 | 40.10 | 78.60 | 93.68 | 85.94 | 31.44 | 91.14 | 89.05 | 74.77 | 44.74 | 82.98 |
| GNNSafe++ | Yes | 91.11 | 82.69 | 53.58 | 76.90 | 94.51 | 89.03 | 32.90 | 76.50 | 92.45 | 82.13 | 35.40 | 90.51 | 92.69 | 84.62 | 40.63 | 81.30 |
| **DeGEM** | No | **99.93** | **99.81** | **0.33** | **84.20** | **99.98** | **99.93** | **0.15** | **83.20** | **97.58** | **93.61** | **12.27** | **93.04** | **99.16** | **97.78** | **4.25** | **86.81** |

**Twitch**

| | OOD Expo | ES | | | FR | | | RU | | | Avg | | | Acc↑ |
|---|---|---|---|---|---|---|---|---|---|---|---|---|---|---|
| | | AUC↑ | AUPR↑ | FPR95↓ | AUC↑ | AUPR↑ | FPR95↓ | AUC↑ | AUPR↑ | FPR95↓ | AUC↑ | AUPR↑ | FPR95↓ | |
| Label Rate = 10% | | | | | | | | | | | | | | |
| OE | Yes | 67.85 | 76.78 | 90.12 | 61.77 | 66.67 | 92.78 | 60.45 | 72.32 | 92.70 | 63.36 | 71.92 | 91.87 | 62.55 |
| Energy-FT | Yes | 85.09 | 90.89 | **73.86** | 85.98 | 89.06 | 76.28 | 87.36 | 91.66 | **62.46** | 86.14 | 90.54 | 70.87 | 63.54 |
| GNNSafe++ | Yes | 75.41 | 82.92 | 84.90 | 76.58 | 82.00 | 87.79 | 81.24 | 90.11 | 80.23 | 77.75 | 85.01 | 84.30 | 63.04 |
| **DeGEM** | No | **93.63** | **97.09** | 73.97 | **96.21** | **97.93** | **17.69** | **94.15** | **97.39** | 62.58 | **94.67** | **97.47** | **51.41** | 60.29 |
| Label Rate = 50% | | | | | | | | | | | | | | |
| OE | Yes | 68.98 | 80.25 | 89.82 | 50.07 | 56.44 | 95.50 | 55.39 | 69.81 | 94.82 | 58.15 | 68.83 | 93.38 | 62.18 |
| Energy-FT | Yes | 90.74 | 95.11 | 63.21 | 73.73 | 78.69 | 84.90 | 97.43 | 98.37 | 11.15 | 87.30 | 90.72 | 53.09 | 64.43 |
| GNNSafe++ | Yes | 94.93 | **97.52** | 40.90 | 86.03 | 91.24 | 80.80 | **99.30** | **99.65** | **1.62** | 93.42 | 96.14 | **41.11** | 64.04 |
| **DeGEM** | No | 93.66 | 97.07 | 65.77 | **96.50** | **98.05** | **16.58** | 92.46 | 96.51 | 66.18 | **94.21** | **97.21** | 49.51 | 61.84 |
| Label Rate = 100% | | | | | | | | | | | | | | |
| OE | Yes | 64.52 | 77.11 | 88.47 | 52.99 | 62.00 | 92.81 | 69.64 | 81.48 | 84.01 | 62.38 | 73.53 | 88.43 | 64.71 |
| Energy-FT | Yes | 93.55 | 96.62 | 42.86 | 70.89 | 75.99 | 87.47 | 97.71 | 98.69 | 10.58 | 87.38 | 90.43 | 46.97 | 63.82 |
| GNNSafe++ | Yes | **97.75** | **98.93** | **9.70** | 92.11 | 95.22 | 64.57 | **99.87** | **99.94** | **0.00** | **96.58** | **98.03** | 24.76 | 65.83 |
| **DeGEM** | No | 94.83 | 97.63 | 43.80 | **97.36** | **98.58** | **2.53** | 94.76 | 97.57 | 41.00 | 95.65 | 97.93 | **29.11** | 64.51 |

**Chameleon**

| | OOD Expo | Structure | | | | Feature | | | | Label | | | | Avg | | | |
|---|---|---|---|---|---|---|---|---|---|---|---|---|---|---|---|---|---|
| | | AUC↑ | AUPR↑ | FPR95↓ | Acc↑ | AUC↑ | AUPR↑ | FPR95↓ | Acc↑ | AUC↑ | AUPR↑ | FPR95↓ | Acc↑ | AUC↑ | AUPR↑ | FPR95↓ | Acc↑ |
| Label Rate = 10% | | | | | | | | | | | | | | | | | |
| OE | Yes | 75.16 | 82.02 | 100.00 | 25.23 | 65.42 | 64.01 | 85.77 | 24.79 | 61.29 | 77.96 | 81.58 | 31.77 | 67.29 | 74.66 | 89.12 | 62.55 |
| Energy-FT | Yes | 55.58 | 67.05 | 100.00 | 24.96 | 45.36 | 43.21 | 97.80 | 30.01 | 67.84 | 78.06 | 59.21 | 39.12 | 56.26 | 62.77 | 85.67 | 63.54 |
| GNNSafe++ | Yes | 24.66 | 47.47 | 100.00 | 30.83 | 50.77 | 48.11 | 95.83 | 28.25 | 50.39 | 69.30 | 86.62 | 41.05 | 41.94 | 54.96 | 94.15 | 63.04 |
| **DeGEM** | No | **99.78** | **99.82** | **0.13** | **43.12** | **99.75** | **99.63** | **1.27** | **40.59** | **90.38** | **95.35** | **41.01** | **48.30** | **96.64** | **98.27** | **14.14** | 60.29 |
| Label Rate = 50% | | | | | | | | | | | | | | | | | |
| OE | Yes | 94.14 | 95.83 | 53.67 | 33.52 | 67.73 | 65.30 | 84.23 | 34.01 | 67.10 | 77.95 | 64.47 | 38.57 | 76.32 | 79.70 | 67.46 | 62.18 |
| Energy-FT | Yes | 76.88 | 83.39 | 100.00 | 29.95 | 55.29 | 55.24 | 95.39 | 31.54 | 73.62 | 83.80 | 70.39 | 39.94 | 68.60 | 74.14 | 88.59 | 64.43 |
| GNNSafe++ | Yes | 31.95 | 54.80 | 100.00 | 36.86 | 55.79 | 51.99 | 94.20 | 34.12 | 50.47 | 72.90 | 83.99 | 38.48 | 46.07 | 59.89 | 92.73 | 64.04 |
| **DeGEM** | No | **99.98** | **99.97** | **0.09** | **53.48** | **99.64** | **99.48** | **1.71** | **48.71** | **91.47** | **95.42** | **37.50** | **55.83** | **97.03** | **98.29** | **13.10** | 61.84 |
| Label Rate = 100% | | | | | | | | | | | | | | | | | |
| OE | Yes | 98.67 | 99.83 | 3.60 | 34.89 | 69.00 | 65.65 | 82.26 | 35.49 | 75.54 | 84.02 | 70.83 | 40.22 | 81.07 | 82.83 | 52.23 | 64.71 |
| Energy-FT | Yes | 98.95 | 99.15 | 0.61 | 38.01 | 65.19 | 64.56 | 94.51 | 38.89 | 75.53 | 84.45 | 66.89 | 39.30 | 79.89 | 82.72 | 54.00 | 63.82 |
| GNNSafe++ | Yes | 36.23 | 58.23 | 100.00 | 36.59 | 57.80 | 58.51 | 93.94 | 37.41 | 50.65 | 73.38 | 87.06 | 42.33 | 48.23 | 63.37 | 93.67 | 65.83 |
| **DeGEM** | No | **99.99** | **99.98** | **0.00** | **57.82** | **99.70** | **99.51** | **1.10** | **57.93** | **89.68** | **94.62** | **39.04** | **64.46** | **96.46** | **98.04** | **13.38** | 64.51 |

**Cornell**

| | OOD Expo | Structure | | | | Feature | | | | Label | | | | Avg | | | |
|---|---|---|---|---|---|---|---|---|---|---|---|---|---|---|---|---|---|
| | | AUC↑ | AUPR↑ | FPR95↓ | Acc↑ | AUC↑ | AUPR↑ | FPR95↓ | Acc↑ | AUC↑ | AUPR↑ | FPR95↓ | Acc↑ | AUC↑ | AUPR↑ | FPR95↓ | Acc↑ |
| Label Rate = 10% | | | | | | | | | | | | | | | | | |
| OE | Yes | 78.38 | 77.47 | 95.08 | 42.86 | 61.03 | 52.40 | 94.54 | 29.93 | 34.76 | 66.40 | 94.74 | **65.71** | 58.06 | 65.42 | 94.78 | 62.55 |
| Energy-FT | Yes | 68.18 | 76.80 | 100.00 | 43.54 | 59.50 | 61.45 | 100.00 | 43.54 | 53.41 | 74.09 | 97.37 | 64.76 | 60.36 | 70.78 | 99.12 | **63.54** |
| GNNSafe++ | Yes | 72.52 | 80.70 | 99.45 | 34.01 | 73.64 | 82.04 | 96.72 | 43.54 | 62.39 | 75.80 | 86.84 | 63.81 | 69.52 | 79.51 | 94.34 | 63.04 |
| **DeGEM** | No | **99.62** | **99.50** | **2.19** | 14.97 | **100.00** | **100.00** | **0.00** | 17.01 | **93.08** | **97.35** | **31.58** | 63.81 | **97.57** | **98.95** | **11.25** | 60.29 |
| Label Rate = 50% | | | | | | | | | | | | | | | | | |
| OE | Yes | 84.13 | 83.96 | 89.07 | **44.22** | 80.47 | 77.97 | 84.15 | 36.05 | 57.22 | 76.85 | 97.37 | 60.95 | 73.94 | 79.59 | 90.20 | 62.18 |
| Energy-FT | Yes | 67.17 | 67.68 | 98.91 | 41.50 | 78.34 | 82.89 | 100.00 | 42.86 | 58.22 | 75.16 | 84.21 | 62.86 | 67.91 | 75.24 | 94.37 | **64.43** |
| GNNSafe++ | Yes | 77.86 | 84.38 | 91.80 | 37.41 | 76.49 | 83.70 | 95.63 | 40.82 | 65.01 | 77.94 | 68.42 | 62.86 | 73.12 | 82.01 | 85.28 | 64.04 |
| **DeGEM** | No | **97.84** | **92.02** | **3.28** | **44.22** | **99.88** | **99.85** | **1.09** | **55.78** | **91.23** | **96.73** | **44.74** | **73.33** | **96.32** | **96.20** | **16.37** | 61.84 |
| Label Rate = 100% | | | | | | | | | | | | | | | | | |
| OE | Yes | 82.05 | 79.76 | 74.32 | 26.53 | 77.29 | 62.74 | 55.74 | 37.41 | 55.79 | 74.93 | 92.11 | 57.14 | 71.71 | 72.47 | 74.05 | 64.71 |
| Energy-FT | Yes | 80.42 | 83.71 | 92.90 | 43.54 | 90.19 | 90.55 | 71.58 | 39.46 | 65.81 | 83.53 | 89.47 | 62.86 | 78.81 | 85.93 | 84.65 | 63.82 |
| GNNSafe++ | Yes | 77.59 | 83.44 | 89.07 | 26.53 | 82.94 | 87.28 | 98.36 | 43.54 | 76.77 | 84.57 | **60.53** | 60.95 | 79.10 | 85.10 | 82.65 | **65.83** |
| **DeGEM** | No | **97.97** | **96.11** | **5.46** | **48.30** | **100.00** | **100.00** | **0.00** | **65.31** | **90.90** | **96.74** | 65.79 | **77.14** | **96.29** | **97.61** | **23.75** | 64.51 |

Table 13: Detailed ablation study results measured by AUC↑ / AUPR↑ / FPR95↓ / Acc↑. 'Eprop' stands for energy propagation. 'MLE-Energy' indicates whether the energy head is trained via maximum likelihood estimation, otherwise, the node energy is obtained by classification logits (Classify-Energy). 'GCL' indicates whether a graph contrastive learning algorithm is employed when training the graph encoder.

**Cora**

| | Eprop | Classify-Energy | MLE-Energy | GCL | MH | CE | ERo | Structure AUROC↑ | AUPR↑ | FPR95↓ | Acc↑ | Feature AUROC↑ | AUPR↑ | FPR95↓ | Acc↑ | Label AUROC↑ | AUPR↑ | FPR95↓ | Acc↑ | Avg AUROC↑ | AUPR↑ | FPR95↓ | Acc↑ |
|---|---|---|---|---|---|---|---|---|---|---|---|---|---|---|---|---|---|---|---|---|---|---|---|
| Energy | | ✓ | | | | | | 79.48 | 54.22 | 66.36 | 79.10 | 89.34 | 79.96 | 55.98 | 79.30 | 93.26 | 82.42 | 33.67 | 90.19 | 87.36 | 72.20 | 52.00 | 82.86 |
| GNNSafe | ✓ | ✓ | | | | | | 87.98 | 79.29 | 76.48 | 75.30 | 92.18 | 86.48 | 49.41 | 75.40 | 92.36 | 81.55 | 34.48 | 88.92 | 90.84 | 82.44 | 53.46 | 79.87 |
| | | | ✓ | | | | | 85.90 | 61.26 | 39.00 | 55.60 | 92.98 | 74.66 | 23.45 | 73.50 | 70.52 | 41.03 | 80.43 | 86.71 | 83.13 | 58.98 | 47.62 | 71.94 |
| | | | | ✓ | | | | 59.46 | 34.47 | 94.13 | 73.20 | 67.92 | 44.80 | 86.60 | 73.30 | 79.94 | 60.09 | 77.38 | 85.44 | 69.11 | 46.45 | 86.04 | 77.31 |
| | | | | ✓ | ✓ | | | 86.32 | 59.53 | 40.77 | 58.10 | 87.54 | 69.55 | 40.21 | 74.70 | 92.75 | 80.01 | 33.57 | 89.56 | 88.87 | 69.70 | 38.18 | 74.12 |
| | | | | ✓ | ✓ | ✓ | | 81.01 | 52.82 | 58.46 | 73.90 | 92.15 | 77.50 | 29.87 | 75.50 | 93.31 | 83.26 | 36.00 | 85.76 | 88.82 | 71.19 | 41.44 | 78.39 |
| | | | | ✓ | ✓ | | ✓ | 86.52 | 62.21 | 43.61 | 75.60 | 96.43 | 84.84 | 9.19 | 78.40 | 94.07 | 84.54 | 26.98 | 87.97 | 92.34 | 77.20 | 26.59 | 80.66 |
| | | | | ✓ | ✓ | ✓ | ✓ | 87.89 | 63.84 | 41.77 | 76.70 | 96.43 | 84.84 | 9.19 | 78.40 | 94.07 | 84.54 | 26.98 | 87.97 | 92.80 | 77.74 | 25.98 | 81.02 |
| | | | | ✓ | ✓ | ✓ | | 98.73 | 96.62 | 5.95 | 82.10 | 97.94 | 92.64 | 6.79 | 80.80 | 95.75 | 90.06 | 22.01 | 92.09 | 97.47 | 93.11 | 11.58 | 85.00 |
| | | | | ✓ | ✓ | ✓ | ✓ | 98.12 | 94.83 | 5.87 | 81.10 | 99.06 | 96.62 | 2.73 | 82.70 | 96.93 | 91.45 | 12.98 | 92.41 | 98.03 | 94.30 | 7.20 | 85.40 |
| | | | | ✓ | ✓ | ✓ | ✓ | 95.22 | 82.38 | 13.04 | 83.50 | 98.84 | 96.44 | 5.21 | 82.80 | 94.65 | 86.29 | 23.63 | 93.04 | 96.24 | 88.37 | 13.96 | 86.45 |
| DeGEM | | | ✓ | ✓ | ✓ | ✓ | ✓ | **99.93** | **99.81** | **0.33** | 84.20 | **99.84** | **99.50** | **0.66** | 84.30 | **97.58** | **93.61** | 12.27 | 93.04 | **99.12** | **97.64** | **4.42** | 87.18 |

**Twitch**

| | Eprop | Classify-Energy | MLE-Energy | GCL | MH | CE | ERo | ES AUROC↑ | AUPR↑ | FPR95↓ | FR AUROC↑ | AUPR↑ | FPR95↓ | RU AUROC↑ | AUPR↑ | FPR95↓ | Avg AUROC↑ | AUPR↑ | FPR95↓ | Acc↑ |
|---|---|---|---|---|---|---|---|---|---|---|---|---|---|---|---|---|---|---|---|---|
| Energy | | ✓ | | | | | | 58.42 | 68.55 | 90.45 | 72.91 | 75.26 | 80.48 | 69.90 | 79.26 | 86.11 | 67.07 | 74.36 | 85.68 | 65.59 |
| GNNSafe | ✓ | ✓ | | | | | | 51.00 | 57.41 | 80.79 | 79.08 | 81.54 | 68.51 | 82.93 | 86.45 | 57.08 | 71.00 | 75.13 | 68.79 | 66.18 |
| | | | ✓ | | | | | 52.12 | 63.85 | 92.58 | 48.78 | 57.40 | 98.60 | 41.94 | 56.59 | 95.07 | 47.61 | 59.28 | 95.42 | 63.87 |
| | | | | ✓ | | | | 74.52 | 85.07 | 86.14 | 79.01 | 84.93 | 86.84 | 69.76 | 80.70 | 85.45 | 74.43 | 83.57 | 86.15 | 60.29 |
| | | | | ✓ | ✓ | | | 83.40 | 90.59 | 80.51 | 90.79 | 93.87 | 66.36 | 80.11 | 88.21 | 78.72 | 84.77 | 90.89 | 75.20 | 60.29 |
| | | | | ✓ | ✓ | ✓ | | 67.06 | 74.88 | 91.09 | 69.53 | 76.28 | 92.76 | 72.85 | 81.23 | 84.08 | 69.81 | 77.46 | 89.31 | 60.61 |
| | | | | ✓ | ✓ | | ✓ | 82.47 | 88.59 | 75.09 | 87.16 | 90.17 | 72.42 | 79.74 | 86.74 | 73.20 | 83.12 | 88.50 | 73.57 | 60.29 |
| | | | | ✓ | ✓ | ✓ | ✓ | 83.40 | 90.59 | 60.04 | 90.79 | 93.87 | 66.36 | 80.11 | 88.21 | 78.72 | 84.77 | 90.89 | 75.20 | 60.29 |
| | | | | ✓ | ✓ | ✓ | | 91.13 | 95.26 | 70.29 | 94.66 | 98.04 | 10.73 | 93.02 | 96.48 | 53.77 | 93.61 | 96.60 | 44.93 | 60.29 |
| | | | | ✓ | ✓ | ✓ | ✓ | 92.14 | 95.71 | 57.92 | 97.82 | 98.77 | 1.80 | 94.92 | 97.47 | 28.83 | 94.96 | 97.32 | 29.51 | 60.29 |
| | | | | ✓ | ✓ | ✓ | ✓ | 90.32 | 92.75 | 51.55 | 96.68 | 96.78 | 3.02 | 90.22 | 93.02 | 52.86 | 92.41 | 94.18 | 35.81 | 60.29 |
| DeGEM | | | ✓ | ✓ | ✓ | ✓ | ✓ | **94.83** | **97.63** | 43.80 | 96.68 | **98.58** | **2.53** | **94.76** | **97.57** | 41.00 | **95.65** | **97.93** | 29.11 | 64.51 |

**Chameleon**

| | Eprop | Classify-Energy | MLE-Energy | GCL | MH | CE | ERo | Structure AUROC↑ | AUPR↑ | FPR95↓ | Acc↑ | Feature AUROC↑ | AUPR↑ | FPR95↓ | Acc↑ | Label AUROC↑ | AUPR↑ | FPR95↓ | Acc↑ | Avg AUROC↑ | AUPR↑ | FPR95↓ | Acc↑ |
|---|---|---|---|---|---|---|---|---|---|---|---|---|---|---|---|---|---|---|---|---|---|---|---|
| Energy | | ✓ | | | | | | 91.94 | 95.28 | 100.00 | 37.14 | 67.75 | 66.73 | 84.15 | 59.92 | 75.25 | 70.18 | 41.69 | 65.75 | 73.20 | 79.09 | 84.77 | 65.75 |
| GNNSafe | ✓ | ✓ | | | | | | 34.36 | 56.86 | 100.00 | 35.33 | 57.46 | 58.24 | 95.83 | 38.07 | 52.18 | 73.72 | 85.09 | 43.43 | 48.00 | 62.94 | 93.64 | 66.18 |
| | | | ✓ | | | | | 89.44 | 91.15 | 86.69 | 33.52 | 85.25 | 83.33 | 68.73 | 42.79 | 72.30 | 83.17 | 76.32 | 60.15 | 82.33 | 85.88 | 77.25 | 63.87 |
| | | | | ✓ | | | | 76.55 | 76.82 | 87.89 | 55.07 | 75.38 | 70.45 | 92.62 | 55.73 | 80.59 | 88.87 | 73.68 | 62.63 | 77.51 | 78.71 | 88.07 | 60.29 |
| | | | | ✓ | ✓ | | | 94.83 | 95.17 | 20.55 | 52.93 | 90.21 | 87.54 | 87.35 | 56.99 | 86.85 | 93.05 | 55.04 | 63.18 | 90.63 | 91.92 | 54.32 | 60.29 |
| | | | | ✓ | ✓ | ✓ | | 97.52 | 96.00 | 4.61 | 51.45 | 95.02 | 92.65 | 21.34 | 57.43 | 85.17 | 92.17 | 52.19 | 63.54 | 92.57 | 93.61 | 26.05 | 60.29 |
| | | | | ✓ | ✓ | | ✓ | 98.16 | 97.74 | 8.30 | 55.13 | 95.69 | 94.68 | 19.41 | 56.56 | 86.86 | 93.37 | 52.41 | 62.72 | 93.57 | 95.26 | 26.71 | 60.29 |
| | | | | ✓ | ✓ | ✓ | ✓ | 98.15 | 97.19 | 6.94 | 56.34 | 97.61 | 94.34 | 5.27 | 56.61 | 87.63 | 93.93 | 48.03 | 64.37 | 94.46 | 95.15 | 20.08 | 60.29 |
| | | | | ✓ | ✓ | ✓ | | 99.16 | 98.27 | 3.34 | 55.46 | 96.97 | 95.83 | 12.78 | 55.84 | 89.56 | 94.63 | 41.89 | 62.08 | 95.23 | 96.24 | 19.33 | 60.29 |
| | | | | ✓ | ✓ | ✓ | ✓ | 99.94 | 99.93 | 0.04 | 54.85 | 98.99 | 98.06 | 3.78 | 55.46 | 89.79 | 94.99 | 38.60 | 64.65 | 96.24 | 97.87 | 14.14 | 60.29 |
| | | | | ✓ | ✓ | ✓ | ✓ | 99.85 | 99.80 | 0.92 | 56.72 | 99.99 | 99.99 | 0.00 | 56.17 | 87.55 | 92.70 | 48.03 | 60.79 | 95.80 | 97.50 | 16.32 | 60.29 |
| DeGEM | | | ✓ | ✓ | ✓ | ✓ | ✓ | **99.99** | **99.98** | **0.00** | 57.82 | 99.70 | 99.51 | 1.10 | 57.93 | 89.68 | 94.62 | 39.04 | 64.46 | 96.46 | 98.04 | 13.38 | 64.51 |

**Cornell**

| | Eprop | Classify-Energy | MLE-Energy | GCL | MH | CE | ERo | Structure AUROC↑ | AUPR↑ | FPR95↓ | Acc↑ | Feature AUROC↑ | AUPR↑ | FPR95↓ | Acc↑ | Label AUROC↑ | AUPR↑ | FPR95↓ | Acc↑ | Avg AUROC↑ | AUPR↑ | FPR95↓ | Acc↑ |
|---|---|---|---|---|---|---|---|---|---|---|---|---|---|---|---|---|---|---|---|---|---|---|---|
| Energy | | ✓ | | | | | | 83.09 | 84.90 | 86.34 | 38.10 | 86.70 | 85.61 | 83.06 | 38.10 | 69.70 | 85.99 | 100.00 | 62.86 | 79.83 | 85.50 | 89.80 | 65.75 |
| GNNSafe | ✓ | ✓ | | | | | | 74.66 | 82.14 | 93.44 | 25.17 | 76.22 | 83.44 | 88.52 | 41.50 | 68.17 | 80.87 | 68.42 | 63.81 | 73.02 | 82.15 | 83.46 | 66.18 |
| | | | ✓ | | | | | 85.24 | 73.90 | 32.79 | 44.22 | 85.95 | 80.87 | 70.49 | 51.70 | 76.44 | 87.65 | 94.74 | 65.71 | 82.54 | 80.80 | 66.01 | 63.87 |
| | | | | ✓ | | | | 63.55 | 65.61 | 92.90 | 42.86 | 67.71 | 59.90 | 95.63 | 48.98 | 72.46 | 87.37 | 76.32 | 63.81 | 67.91 | 70.96 | 88.28 | 60.29 |
| | | | | ✓ | ✓ | | | 93.64 | 91.96 | 20.22 | 44.22 | 89.50 | 86.49 | 60.11 | 49.66 | 77.87 | 90.31 | 97.37 | 65.71 | 87.00 | 89.58 | 59.23 | 60.29 |
| | | | | ✓ | ✓ | ✓ | | 91.06 | 87.09 | 27.32 | 40.82 | 88.47 | 86.19 | 36.07 | 51.02 | 75.31 | 88.31 | 68.42 | 63.81 | 84.95 | 87.20 | 43.94 | 60.29 |
| | | | | ✓ | ✓ | | ✓ | 89.81 | 86.46 | 30.40 | 44.22 | 91.66 | 85.79 | 24.04 | 51.02 | 77.44 | 89.40 | 76.32 | 65.71 | 86.31 | 87.22 | 43.65 | 60.29 |
| | | | | ✓ | ✓ | ✓ | ✓ | 87.37 | 84.68 | 51.37 | 45.58 | 93.88 | 90.46 | 27.32 | 50.34 | 74.81 | 89.05 | 97.37 | 67.62 | 85.35 | 88.06 | 58.69 | 60.29 |
| | | | | ✓ | ✓ | ✓ | | 99.09 | 97.14 | 3.83 | 43.54 | 97.50 | 96.09 | 13.11 | 62.59 | 91.13 | 96.67 | 57.89 | 71.43 | 95.90 | 96.63 | 24.94 | 60.29 |
| | | | | ✓ | ✓ | ✓ | ✓ | 98.54 | 97.96 | 5.46 | 48.30 | 99.56 | 99.36 | 2.19 | 61.90 | 88.37 | 95.62 | 52.63 | 68.57 | 95.49 | 97.65 | 20.09 | 60.29 |
| | | | | ✓ | ✓ | ✓ | ✓ | 98.46 | 95.55 | 2.73 | 45.58 | 99.51 | 99.28 | 1.64 | 64.63 | 83.98 | 93.97 | 78.95 | 70.48 | 93.99 | 96.27 | 27.77 | 60.29 |
| DeGEM | | | ✓ | ✓ | ✓ | ✓ | ✓ | 97.97 | 96.11 | 5.46 | 48.30 | **100.00** | **100.00** | **0.00** | 65.31 | 90.90 | 96.74 | 65.79 | 77.14 | 96.29 | 97.61 | 23.75 | 64.51 |

Table 14: The detailed OOD detection performance of different GCL algorithms, measured by AUC↑ / AUPR↑ / FPR95↓.

**Cora**

| | Structure AUC↑ | AUPR↑ | FPR95↓ | Feature AUC↑ | AUPR↑ | FPR95↓ | Label AUC↑ | AUPR↑ | FPR95↓ | Avg AUC↑ | AUPR↑ | FPR95↓ |
|---|---|---|---|---|---|---|---|---|---|---|---|---|
| GCN | 59.54 | 37.00 | 94.83 | 70.85 | 46.21 | 81.46 | 74.37 | 49.15 | 86.71 | 68.25 | 44.12 | 87.67 |
| GRACE | 67.56 | 42.24 | 90.69 | **89.71** | **78.84** | _49.00_ | _91.75_ | _74.81_ | **32.56** | _83.01_ | _65.29_ | _57.42_ |
| SUGRL | _74.97_ | _45.75_ | _67.98_ | 84.52 | _71.12_ | 63.04 | 87.73 | 72.97 | 45.33 | 82.41 | 63.28 | 58.78 |
| DGI | **86.32** | **59.53** | **40.77** | _87.54_ | 69.55 | **40.21** | **92.75** | **80.01** | _33.57_ | **88.87** | **69.70** | **38.18** |

**Twitch**

| | ES AUC↑ | AUPR↑ | FPR95↓ | FR AUC↑ | AUPR↑ | FPR95↓ | RU AUC↑ | AUPR↑ | FPR95↓ | Avg AUC↑ | AUPR↑ | FPR95↓ |
|---|---|---|---|---|---|---|---|---|---|---|---|---|
| GCN | 50.89 | 64.14 | 94.49 | 53.82 | 57.39 | 94.23 | 57.10 | _73.16_ | 95.30 | 53.94 | 64.90 | 94.67 |
| GRACE | 67.66 | 73.59 | 88.15 | 63.54 | 64.71 | 93.67 | _66.15_ | 72.59 | _86.64_ | 65.78 | 70.30 | 89.48 |
| SUGRL | _70.89_ | _80.04_ | _84.53_ | _84.23_ | _87.11_ | _74.77_ | 61.32 | 75.39 | 94.07 | _72.15_ | _79.90_ | _84.46_ |
| DGI | **83.40** | **90.59** | **80.51** | **90.79** | **93.87** | **66.36** | **80.11** | **88.21** | **78.72** | **84.77** | **90.89** | **75.20** |

**Chameleon**

| | Structure AUC↑ | AUPR↑ | FPR95↓ | Feature AUC↑ | AUPR↑ | FPR95↓ | Label AUC↑ | AUPR↑ | FPR95↓ | Avg AUC↑ | AUPR↑ | FPR95↓ |
|---|---|---|---|---|---|---|---|---|---|---|---|---|
| GCN | 68.19 | 73.99 | 99.78 | 84.64 | 79.36 | **56.92** | 71.13 | 82.73 | 93.86 | 74.65 | 78.69 | 83.52 |
| GRACE | _89.95_ | _90.68_ | _60.52_ | _87.63_ | **87.94** | 88.45 | _86.38_ | _92.23_ | 60.09 | _87.99_ | _90.28_ | 69.69 |
| SUGRL | 85.91 | 84.02 | 62.71 | 78.67 | 71.97 | _66.45_ | 83.56 | 90.11 | 61.84 | 82.71 | 82.03 | _63.67_ |
| DGI | **94.83** | **95.17** | **20.55** | **90.21** | _87.54_ | 87.35 | **86.85** | **93.05** | **55.04** | **90.63** | **91.92** | **54.32** |

**Cornell**

| | Structure AUC↑ | AUPR↑ | FPR95↓ | Feature AUC↑ | AUPR↑ | FPR95↓ | Label AUC↑ | AUPR↑ | FPR95↓ | Avg AUC↑ | AUPR↑ | FPR95↓ |
|---|---|---|---|---|---|---|---|---|---|---|---|---|
| GCN | 77.25 | 83.09 | 100.00 | 73.54 | 70.99 | 93.44 | 76.47 | _90.65_ | 86.84 | 75.75 | 81.58 | 93.43 |
| GRACE | _84.94_ | _86.86_ | _74.86_ | _80.39_ | _78.44_ | _77.60_ | _76.94_ | **91.05** | 86.84 | _80.76_ | _85.45_ | _79.77_ |
| SUGRL | 60.43 | 55.31 | 96.17 | 62.09 | 54.02 | 91.26 | 68.51 | 84.73 | 86.84 | 63.67 | 64.69 | 91.42 |
| DGI | **93.64** | **91.96** | **20.22** | **89.50** | **86.49** | **60.11** | **77.87** | 90.31 | 97.37 | **87.00** | **89.58** | **59.23** |

