# OpenReview forum: "Decoupled Graph Energy-based Model for Node Out-of-Distribution Detection on Heterophilic Graphs"
_ICLR.cc/2025/Conference — ICLR 2025 Poster_

### Official Review · Reviewer_fY4f · 2024-10-28

**Soundness:** 3
**Presentation:** 2
**Contribution:** 2
**Rating:** 6
**Confidence:** 3

**Summary:**

This work proposes a method named DeGEM for graph OOD node detection, which can overcome the heterophily issue and the computational challenges.

**Strengths:**

- This work seems to be theoretically solid and technically sound.
- Extensive experiments show that the proposed method has a promising performance.
- The authors provided the source code in Supplementary Material, facilitating good reproducibility of this work.

**Weaknesses:**

- In Abstract, the authors argue that GNNSafe has significant performance degradation on heterophilic graphs. However, there is a recent work [1] that has similar consideration.
- In Introduction, the authors only analyze the limitation of one recent graph OOD method named GNNSafe. However, as discussed in Section 5, there are many existing graph OOD methods. What are the fundamental research challenges that cannot be addressed by these existing methods?
- There are some minors. For example, the text font in tables is too small. It would be better to provide all the source scripts as well as the used datasets and give the anonymous repo link in the text.


Refs:

[1] Ren, Lingfei, Ruimin Hu, Zheng Wang, Yilin Xiao, Dengshi Li, Junhang Wu, Jinzhang Hu, Yilong Zang, and Zijun Huang. "Heterophilic Graph Invariant Learning for Out-of-Distribution of Fraud Detection." In ACM Multimedia 2024. 2024.

**Questions:**

Please see the weaknesses listed above.

---

> ### Author Response · Authors · 2024-11-25
>
> Thank you for your time and effort in reviewing our paper. We very much appreciate your insightful comments and your recognition of our work.  We hereby address the concerns below.
>
>
> > In Abstract, the authors argue that GNNSafe has significant performance degradation on heterophilic graphs. However, there is a recent work [1] that has similar consideration.
> >
> > [1] Ren, Lingfei, Ruimin Hu, Zheng Wang, Yilin Xiao, Dengshi Li, Junhang Wu, Jinzhang Hu, Yilong Zang, and Zijun Huang. "Heterophilic Graph Invariant Learning for Out-of-Distribution of Fraud Detection." In ACM Multimedia 2024. 2024.
>
> We agree with [1] that also considered heterophilic graphs. However, their work differs significantly from ours - they focused on **OOD generalization** in the context of fraud detection, while we focus on **node-level OOD detection**.
>
> > In Introduction, the authors only analyze the limitation of one recent graph OOD method named GNNSafe. However, as discussed in Section 5, there are many existing graph OOD methods. What are the fundamental research challenges that cannot be addressed by these existing methods?
>
> The fundamentally unaddressed issue for existing graph-based node ood detection methods is that - they rely on homophily assumption. This makes existing works show serious performance degradation in heterophilic graphs. In contrast, our work develops DeGEM showing state-of-the-art performance on both homophilic and heterophilic graphs. In particular, we discuss graph OOD detection along two distinct lines in Related Work (Section 5): 1) graph-level OOD detection ; 2) node-level OOD detection. Among these, the former has received more research attention. Moreover, graph-level OOD detection differs significantly from node-level OOD detection -- in graph-level detection, different graphs can be viewed as following i.i.d. distribution, whereas in node-level settings, nodes are interdependent. This interdependency presents a fundamental and unique challenge for node-level OOD detection, hindering straightforward adaptation of existing methods in other domains. A few works have been proposed to tackle the node interdependencies in node OOD detection. However, all of the existing graph-based methods rely on homophily assumption which cannot deal with heterophily issues for node-level OOD detection.
>
> > There are some minors. For example, the text font in tables is too small. It would be better to provide all the source scripts as well as the used datasets and give the anonymous repo link in the text.
>
> Thanks for your suggestion. We will improve the readability of the table in the revision.
> We follow your advice of providing our code using an anonymous Github link: https://anonymous.4open.science/r/DeGEM\_ICLR2025\_rebuttal-B801/README.md, and we have attached the anonymous link in our abstract.

---

### Official Review · Reviewer_JnX7 · 2024-11-01

**Soundness:** 3
**Presentation:** 3
**Contribution:** 3
**Rating:** 6
**Confidence:** 2

**Summary:**

The paper proposes a Decoupled Graph Energy-based Model (DeGEM) for detecting out-of-distribution (OOD) nodes on graphs, specifically addressing challenges on heterophilic graphs where existing models struggle. Extensive experiments validate DeGEM's superior performance, highlighting its robustness and scalability.

**Strengths:**

1. The paper is easy to follow.

2. OOD node detection is an important topic in the graph machine learning literature.

3. The model design is technically novel.

4. The performance of the proposed model surpasses that of the existing baseline methods.

**Weaknesses:**

1. More detailed explanations are needed regarding the state-of-the-art performance on homophilic graph datasets. The paper introduces specific components designed to address the heterophily phenomenon in graph datasets, which lead to their effectiveness on heterophilic datasets. However, it remains unclear why these components are also beneficial for homophilic data. Providing a rationale or theoretical basis for this performance would strengthen the paper's quality.

2. Given the paper’s focus on heterophilic graphs, it would be helpful to include experiments that analyze how varying levels of heterophily affect model performance. This could be similar to Figure 2 in [1], which explores performance across different heterophily levels, offering valuable insights into the model's adaptability.

[1] Zhu, Jiong, et al. "Graph neural networks with heterophily." Proceedings of the AAAI conference on artificial intelligence. Vol. 35. No. 12. 2021.

**Questions:**

It appears that the uncertainty estimation methods proposed in previous GNN literature [1, 2] could also be applied for OOD node detection. Would it be possible for the authors to consider including these methods as baselines?

[1] Huang, Kexin, et al. "Uncertainty quantification over graph with conformalized graph neural networks." Advances in Neural Information Processing Systems 36.

[2] Hart, Russell, et al. "Improvements on Uncertainty Quantification for Node Classification via Distance Based Regularization." Advances in Neural Information Processing Systems 36.

---

> ### Author Response · Authors · 2024-11-25
>
> Thank you for your time and effort in reviewing our paper. We very much appreciate your insightful comments and your recognition of our work.  We hereby address the concerns below.
>
> > More detailed explanations are needed regarding the state-of-the-art performance on homophilic graph datasets. The paper introduces specific components designed to address the heterophily phenomenon in graph datasets, which lead to their effectiveness on heterophilic datasets. However, it remains unclear why these components are also beneficial for homophilic data. Providing a rationale or theoretical basis for this performance would strengthen the paper's quality.
>
> We clarify that the components we have proposed are applicable to general graphs, including both heterophilic and homophilic graphs. Our focus is on heterophilic graphs, where previous work has performed poorly, leaving substantial room for improvement. Specifically, we suggest training Energy-based Models (EBMs) by Maximum Likelihood Estimation (MLE), this can enhance the modeling ability of any data distribution for constructing better OOD detection. Moreover, the proposed MH Graph Encoder can adaptively extract both local and global information. In homophilic graphs, the learned model will focus more on local neighbor nodes. However, in heterophilic graphs, the learned model extracts more useful information from distant but similar nodes. Thus it improves the learned representation in both homophilic and heterophilic graphs.
>
> > Given the paper’s focus on heterophilic graphs, it would be helpful to include experiments that analyze how varying levels of heterophily affect model performance. This could be similar to Figure 2 in [1], which explores performance across different heterophily levels, offering valuable insights into the model's adaptability.
> >
> > [1] Zhu, Jiong, et al. "Graph neural networks with heterophily." Proceedings of the AAAI conference on artificial intelligence. Vol. 35. No. 12. 2021.
>
> Following your advice, we conduct additional experiments on varying the homophilic ratio. The synthetic graphs are constructed following [1]. We report the AUROC results on synthetic-Cora below, and we use the Feature OOD setting.
>
> | homo ratio |  0.0  |  0.2  |  0.4  |  0.6  |  0.8  |  1.0  |
> |------------|:------:|:------:|:------:|:------:|:------:|:------:|
> | GNNSafe    | 69.02  | 70.65  | 72.73  | 73.72  | 79.08  | 79.12  |
> | DeGEM      | 99.66  | 96.56  | 98.39  | 99.11  | 99.86  | 99.96  |
>
> As the homophilic ratio decreases from 1.0 to 0.0, the performance of GNNSafe drops significantly from 79.12 to 69.02. In contrast, DeGEM consistently maintains a high performance across homophilic ratios, achieving around 99 in most cases. This indicates that GNNSafe struggles to handle heterophily, whereas our proposed method demonstrates strong performance on both homophilic and heterophilic graphs.

---

> ### Author Response · Authors · 2024-11-25
> **Follow-up Response**
>
> > It appears that the uncertainty estimation methods proposed in previous GNN literature [1, 2] could also be applied for OOD node detection. Would it be possible for the authors to consider including these methods as baselines?
> >
> > [1] Huang, Kexin, et al. "Uncertainty quantification over graph with conformalized graph neural networks." Advances in Neural Information Processing Systems 36.
> >
> > [2] Hart, Russell, et al. "Improvements on Uncertainty Quantification for Node Classification via Distance Based Regularization." Advances in Neural Information Processing Systems 36.
>
> Yes, uncertainty methods can also be used for node OOD detection. In our paper, we have included two uncertainty-based baselines: GKDE and GPN. We are glad to provide additional uncertainty baselines. Due to time limitations, we conduct experiments using [2] as an additional baseline, which is a stronger variant based on GPN. And the results for homophilic/heterophilic graphs are shown below respectively.
>
> |  Method |    Cora   |        |         |        |        |        | Amazon |        |         |        |        |        | Twitch |        |        |        |  Arxiv |        |        |        |   Avg  |        |
> |:-------:|:---------:|:------:|:-------:|:------:|:------:|:------:|:------------:|:------:|:-------:|:------:|:------:|:------:|:------:|:------:|:------:|:------:|:------:|:------:|:------:|:------:|:------:|:------:|
> |         | Structure |        | Feature |        |  Label |        |   Structure  |        | Feature |        |  Label |        |   ES   |   FR   |   RU   |  Acc↑  |  2018  |  2019  |  2020  |  Acc↑  |        |        |
> |         |    AUC↑   |  Acc↑  |   AUC↑  |  Acc↑  |  AUC↑  |  Acc↑  |     AUC↑     |  Acc↑  |   AUC↑  |  Acc↑  |  AUC↑  |  Acc↑  |  AUC↑  |  AUC↑  |  AUC↑  |        |  AUC↑  |  AUC↑  |  AUC↑  |        |  AUC↑  |  Acc↑  |
> | \[2\]     |   82.21   | 81.00  |  88.06  | 78.80  | 91.73  | 91.77  |    97.00     | 79.38  |  86.65  | 63.84  | 94.99  | 91.00  | 84.29  | 76.79  | 78.20  | 60.29  |   OOM  |   OOM  |   OOM  |   OOM  |   OOM  |   OOM  |
> | GNNSafe |   87.98   | 75.30  |  92.18  | 75.40  | 92.36  | 88.92  |    98.69     | 93.74  |  98.47  | 92.96  | 97.34  | 95.72  | 51.00  | 79.08  | 82.93  | 66.18  | 67.27  | 69.20  | 79.02  | 54.26  | 88.73  | 80.31  |
> | DeGEM   |   99.93   | 84.20  |  99.84  | 84.30  | 97.58  | 93.04  |    100.00    | 94.49  |  99.91  | 93.97  | 99.28  | 95.80  | 94.83  | 97.36  | 94.76  | 64.51  | 81.30  | 86.00  | 86.01  | 58.20  | 97.08  | 83.56  |
>
> |  Method | Chameleon |        |         |        |        |        |   Actor   |        |         |        |         |        |  Cornell  |        |         |        |        |        |   Avg  |        |
> |:-------:|:---------:|:------:|:-------:|:------:|:------:|:------:|:---------:|:------:|:-------:|:------:|:-------:|:------:|:---------:|:------:|:-------:|:------:|:------:|:------:|:------:|:------:|
> |         | Structure |        | Feature |        |  Label |        | Structure |        | Feature |        |  Label  |        | Structure |        | Feature |        |  Label |        |        |        |
> |         |    AUC↑   |  Acc↑  |   AUC↑  |  Acc↑  |  AUC↑  |  Acc↑  |    AUC↑   |  Acc↑  |   AUC↑  |  Acc↑  |   AUC↑  |  Acc↑  |    AUC↑   |  Acc↑  |   AUC↑  |  Acc↑  |  AUC↑  |  Acc↑  |  AUC↑  |  Acc↑  |
> | \[2\] |   93.80   | 30.06  |  64.38  | 31.16  | 77.33  | 38.02  |   78.18   | 25.97  |  63.81  | 17.38  |  68.01  | 33.55  |   90.91   | 43.54  |  78.97  | 46.26  | 82.93  | 63.81  | 77.59  | 36.64  |
> | GNNSafe |   34.36   | 35.33  |  57.46  | 38.07  | 52.18  | 43.43  |   31.76   | 26.30  |  50.66  | 26.20  |  51.60  | 37.92  |   74.66   | 25.17  |  76.22  | 41.50  | 68.17  | 63.81  | 55.23  | 37.52  |
> | DeGEM   |   99.99   | 57.82  |  99.70  | 57.93  | 89.68  | 64.46  |   99.76   | 31.97  |  99.98  | 33.87  | 100.00  | 36.02  |   97.97   | 48.30  | 100.00  | 65.31  | 90.90  | 77.14  | 97.55  | 52.53  |
>
> It can be seen that compared to [2], our method achieves consistently better performance across homophilic and heterophilic graphs.

---

> > ### Comment · Reviewer_JnX7 · 2024-11-26
> > **Thank you for the response**
> >
> > Thank you for the response. Based on the replies, I would keep my score

---

> > > ### Author Response · Authors · 2024-11-27
> > >
> > > We thank you for acknowledging our work. Thanks again for your time and effort in reviewing our work.

---

### Official Review · Reviewer_ta9f · 2024-11-02

**Soundness:** 3
**Presentation:** 3
**Contribution:** 3
**Rating:** 8
**Confidence:** 4

**Summary:**

This paper, titled “Decoupled Graph Energy-based Model for Node Out-of-Distribution Detection on Heterophilic Graphs,” presents a new model, DeGEM, aimed at detecting Out-of-Distribution (OOD) data in nodes on heterophilic graphs. By decoupling the graph encoder from the energy head, DeGEM addresses challenges in traditional methods, such as reliance on homophily assumptions and the complexity of MCMC sampling in graph structures. Experimental results show that, even without exposure to OOD data, DeGEM outperforms state-of-the-art models on both homophilic and heterophilic graphs.

**Strengths:**

1.The DeGEM model enhances OOD detection performance on graphs by decoupling the graph encoder and energy head, which helps avoid the performance degradation seen on heterophilic graphs due to homophily assumptions. This design demonstrates strong generalizability.

2.By moving MCMC sampling to the latent space, DeGEM reduces computational complexity, avoiding direct sampling on graph structures and achieving good scalability.

3.The paper conducts extensive experiments on both homophilic and heterophilic graphs, showing performance improvements across different graph types, which illustrates the model’s practical potential.

4.DeGEM enables effective OOD detection without the need for OOD data during training, making it highly valuable for real-world applications where unsupervised adaptability is crucial.

**Weaknesses:**

1.Although the decoupled design reduces dependency on graph structure and improves computational efficiency, the introduction of multiple components (such as GCL, conditional energy, and recurrent updates) adds complexity to the model structure. I suggest that key components be simplified in future versions.

2.While the paper includes some ablation studies, it does not discuss independent contributions from specific components such as the Multi-Hop encoder, Energy Readout, and Conditional Energy modules. Further independent testing of these components is recommended for future research.

**Questions:**

1.Although the decoupled design reduces dependency on graph structure and improves computational efficiency, could the introduction of multiple components (such as GCL, conditional energy, and recurrent updates) make the model structure overly complex? Could future versions simplify key components to reduce implementation difficulty?

2.While the paper includes some ablation studies, why were independent tests not conducted for specific modules like the Multi-Hop encoder, Energy Readout, and Conditional Energy? Could future research add such independent testing to more comprehensively assess the contributions of each component?

---

> ### Author Response · Authors · 2024-11-25
>
> Thank you for your time and effort in reviewing our paper. We very much appreciate your insightful comments and your recognition of our work.  We hereby address the concerns below.
>
> > While the paper includes some ablation studies, it does not discuss independent contributions from specific components such as the Multi-Hop encoder, Energy Readout, and Conditional Energy modules. Further independent testing of these components is recommended for future research.
>
> Thanks for your suggestion. We have evaluated +MH (Row 6 in Table 4), +MH+CE (Row 7), and +MH+ERo (Row 8) in the original paper. We mainly evaluate the effectiveness of each component by adding it one by one. We believe this already could indicate the effectiveness of each component. There are many existing works that also evaluate their components by adding them one by one in the ablation study ([a,b,c,d]).
>
> However, we are also glad to provide additional ablation study of +CE, +ERo, and recurrent update (+CE+ERo). We would like to indicate that the CE and ERo should be used together to construct the recurrent update for enhancing performance. We refer to the model that incorporates DGI and MLE (Maximum Likelihood Estimation) for learning energy based on GCN as the base model. The detailed ablation results are shown below.
>
> |                |       | Components |        |     |    |    |     | Homophily |        |        | Heterophily |         |        |   Avg  |
> |----------------|-------|:----------:|:------:|:---:|:--:|:--:|:---:|:---------:|:------:|:------:|:-----------:|:-------:|:------:|:------:|
> |                |       |  Classify- Energy | MLE-Energy   | GCL | MH | CE | ERo |    Cora   | Twitch |   Avg  |    Cham.    | Cornell |   Avg  |        |
> | Base w/o MLE   |       |      √     |        |  √  |    |    |     |   83.13   | 47.61  | 65.37  |    82.33    |  82.54  | 82.44  | 73.90  |
> | Base           |       |            |    √   |  √  |    |    |     |   88.87   | 84.77  | 86.82  |    90.63    |  87.00  | 88.82  | 87.82  |
> | Base+CE        |       |            |    √   |  √  |    |  √ |     |   88.82   | 69.81  | 79.32  |    92.57    |  84.95  | 88.76  | 84.04  |
> | Base+ERo       |       |            |    √   |  √  |    |    |  √  |   92.34   | 83.12  | 87.73  |    93.57    |  86.31  | 89.94  | 88.84  |
> | Base+CE+ERo    |       |            |    √   |  √  |    |  √ |  √  |   92.80   | 84.77  | 88.78  |    94.46    |  85.35  | 89.91  | 89.35  |
> | Base+MH        |       |            |    √   |  √  |  √ |    |     |   97.47   | 93.61  | 95.54  |    95.23    |  95.90  | 95.57  | 95.55  |
> | Base+MH+CE     |       |            |    √   |  √  |  √ |  √ |     |   98.03   | 94.96  | 96.50  |    96.24    |  95.49  | 95.87  | 96.18  |
> | Base+MH+ERo    |       |            |    √   |  √  |  √ |    |  √  |   96.24   | 92.41  | 94.32  |    95.80    |  93.99  | 94.89  | 94.61  |
> | Base+MH+CE+ERo | **DeGEM** |            |    √   |  √  |  √ |  √ |  √  |   99.12   | 95.65  | 97.38  |    96.46    |  96.29  | 96.37  | 96.88  |
>
>  The results clearly show the effectiveness of the MH graph encoder and Recurrent Update.
>
> [a] LSGNN: Towards General Graph Neural Network in Node Classification by Local Similarity, IJCAL 2023.
>
> [b] Graph Self-supervised Learning with Accurate Discrepancy Learning, NeurIPS 2022.
>
> [c] Analyzing and Improving the Image Quality of StyleGAN, CVPR 2020.
>
> [d] Text Is MASS: Modeling as Stochastic Embedding for Text-Video Retrieval, CVPR 2024.
>
> > Although the decoupled design reduces dependency on graph structure and improves computational efficiency, could the introduction of multiple components (such as GCL, conditional energy, and recurrent updates) make the model structure overly complex? Could future versions simplify key components to reduce implementation difficulty?
>
>
> We will work on simplifying this model in future work. However, we would like to highlight that, these components can be easily implemented in practice. For example, the GCL algorithm we used is plugged into our backbone model without lots of effort.
> Additionally, the design you mentioned does not occupy much running time. For example, from Table 7 in the Appendix, the running time of GCL only contributes to 6\% of total time consumption.

---

> ### Comment · Reviewer_ta9f · 2024-11-27
>
> Thank you for the response. Based on the replies, I would raise my score.

---

> > ### Author Response · Authors · 2024-11-27
> >
> > We thank you for acknowledging our work and for raising the score. Thanks again for your time and effort in reviewing our work.

---

### Official Review · Reviewer_6WFP · 2024-11-07

**Soundness:** 3
**Presentation:** 3
**Contribution:** 3
**Rating:** 6
**Confidence:** 3

**Summary:**

This paper studies out-of-distribution detection on graph data. The authors argue that existing models for this problem derives node-wise energy function that ignores the inter-dependence among node instances for uncertainty modeling and relies on homophily assumption for energy propagation. To address these limitations, the authors propose a new energy-based model as well as a new training scheme that can address graph heterophily and enables effective training of the energy model. Experiments on benchmark datasets show the consistent improvements of the proposed model over state-of-the-arts.

**Strengths:**

1. The paper is well motivated and studies an important problem in graph learning

2. The proposed model seems reasonable and sound

3. The experiment results are promising and solid

**Weaknesses:**

1. The argument that existing works on out-of-distribution detection resort to energy function independent for each node is arguably not true. There is a recent paper [1] that considers Dirichlet energy and extends the node-wise energy-based modeling for accommodating the neighborhood information in the graph.

2. The proposed model is computationally expensive, and more justification on the necessity of the proposed components that complex the model is needed. Particularly, compared to GNNSafe and other peer models, the additional computational cost of the proposed method seems considerably large, that questions the practical efficacy and scalability of the model in large datasets.

[1] Graph Out-of-Distribution Detection Goes Neighborhood Shaping, ICML 2024.

**Questions:**

1. Is there any intuition why the proposed model can address graph heterophily?

2. What is the computational complexity of the training algorithm and how does it compare with the other models?

3. What is the time/memory costs of the model and how does it compare with the other models?

---

> ### Author Response · Authors · 2024-11-25
>
> Thank you for your time and effort in reviewing our paper. We very much appreciate your acknowledgment of our motivation and the strong experiment results. We hereby address the concerns below.
>
> > The argument that existing works on out-of-distribution detection resort to energy function independent for each node is arguably not true. There is a recent paper [1] that considers Dirichlet energy and extends the node-wise energy-based modeling for accommodating the neighborhood information in the graph.
> > [1] Graph Out-of-Distribution Detection Goes Neighborhood Shaping, ICML 2024.
>
> There may be some misunderstanding here. We do not argue there is no work considering node dependence in energy modeling. In contrast, we acknowledge that GNNSafe has considered the node dependence by energy propagation.
> We have revised the text to clarify this distinction more effectively. Hope this makes things clearer. We attach the abstract involving modifications below, with ***italics*** indicating newly added portions:
>
> Despite extensive research efforts focused on Out-of-Distribution (OOD) detection on images, OOD detection on nodes in graph learning remains underexplored. The dependence among graph nodes hinders the trivial adaptation of existing approaches ***on images*** that assume inputs to be i.i.d. sampled, since many unique features and challenges specific to graphs are not considered, such as the heterophily issue. Recently, GNNSafe, ***which considers node dependence***, adapted energy-based detection to the graph domain with state-of-the-art performance.
>
> > Is there any intuition why the proposed model can address graph heterophily?
>
> This is because we abandoned the inductive bias of homophily assumption. Specifically, we removed the energy propagation used in GNNsafe to avoid the severe decrease in performance on heterophilic graphs. Additionally, we introduced MLE to train the Energy function, improving the model's expressive ability so that our model is more powerful in OOD detection; and the Multi-Hop (MH) Graph Encoder can extract information from neighbor nodes to distant nodes. Since similar nodes often lie in the distance in heterophilic graphs, combining both local and global information can bring benefits in addressing heterophilic issues. We also introduced Conditional energy, which allows us to use global and local information in energy calculation, further alleviating heterophilic information. These performance-enhancing components for OOD detection do not rely on the homophily assumption, thus enabling our method to achieve state-of-the-art performance on heterophilic graphs.
>
> > The proposed model is computationally expensive, and more justification on the necessity of the proposed components that complex the model is needed. More justification on the necessity of the proposed components that complex the model is needed.
>
> Overall, our model's AUROC largely outperformed previous graph-based state-of-the-art (GNNSafe) by 26.33% on average among all graphs, while requiring less than 7% additional training time cost on average. In fact, it is common by trading computational cost for performance improvements in the community. For example: 1) There is a well-known technique called Sharpness-Aware Minimization (SAM) [a] that doubles the computational cost at each training iteration while bringing about 1%-2% accuracy improvement in image classification. 2) The famous generative model - Diffusion [b], which has thousands of times higher inference cost in its original formulation compared to the previous SOTA generative model GANs, but achieves superior performance.
>
> [a] Sharpness-Aware Minimization for Efficiently Improving Generalization, ICLR 2021.
>
> [b] Denoising Diffusion Probabilistic Models, NeurIPS 2020.

---

> > ### Author Response · Authors · 2024-11-25
> > **Follow-up Response**
> >
> > > What is the computational complexity of the training algorithm and how does it compare with the other models?
> >
> > Compared to standard GNN classification, our additional computational complexity comes from the introduced GCL and MCMC sampling. We have a similar computational complexity compared to other models. We provide the analysis below.
> >
> > Suppose that GCN is used as the graph encoder in the baselines. Then, the baselines have a complexity of $\mathcal{O}(L m d_0 d_1)$, where $m$ is the number of edges, $d_0$ is the initial node dimension, $d_1$ is the latent dimension of node representation, and $L$ is the number of layers (propagation numbers).
> >
> > In terms of our methods, the GCL processes both positive nodes and negative nodes, but the propagation for positive nodes is conducted in pre-process, with only the transformation remaining in training iterations. So the complexity for GCL is $\mathcal{O}(L n d_0 d_1 + L m d_0 d_1) \approx \mathcal{O}(L m d_0 d_1)$. since normally $m >> n$, where $n$ is the number of nodes. The MCMC sampling has a complexity of $\mathcal{O}((nd_1d_2+nd_2)K)=\mathcal{O}(n d_1 d_2 K)$, where $d_2$ is the hidden dimension of Energy Function ($2$-layer MLP), and $K$ is the number of steps of MCMC sampling.
> >
> > This actually indicates that our method has scalability comparable to GNNs, since the seemingly computationally intensive MCMC steps do not suffer from edges, with complexity only scaling linearly with the number of nodes.
> >
> > > What is the time/memory costs of the model and how does it compare with the other models?
> >
> > Compared to existing baselines, our model demonstrates similar computational costs overall, while achieving significantly better node OOD detection performance. We reevaluate the comparisons of time consumption between baselines and DeGEM, the results are shown below.
> >
> >
> > | Method | Cora | Amazon | Twitch | Arxiv | Chameleon | Actor | Cornell | Avg |
> > |----------|-------|---------|---------|---------|------------|--------|----------|-----|
> > | MSP | 3.95 | 17.98 | 24.45 | 104.16 | 5.87 | 6.35 | 3.16 | 23.70 |
> > | ODIN | 6.43 | 31.65 | 42.72 | 225.78 | 10.23 | 10.68 | 5.38 | 47.55 |
> > | Mahalanobis | 28.77 | 96.66 | 187.42 | 4956.43 | 36.32 | 71.06 | 20.40 | 771.01 |
> > | Energy | 4.81 | 21.40 | 26.36 | 117.02 | 7.14 | 7.78 | 4.10 | 26.94 |
> > | GKDE | 5.21 | 10.58 | 14.15 | 45.06 | 5.37 | 7.72 | 4.48 | 13.22 |
> > | GPN | 15.28 | 27.88 | 23.55 | OOM | 14.39 | 22.51 | 11.43 | 19.17 |
> > | OODGAT | 5.18 | 23.13 | 33.80 | 144.25 | 7.78 | 8.70 | 4.56 | 32.49 |
> > | GNNSafe | 5.59 | 22.28 | 26.42 | 136.55 | 7.55 | 8.12 | 3.83 | 30.05 |
> > | OE | 5.72 | 24.40 | 27.39 | 129.01 | 7.63 | 8.38 | 4.27 | 29.54 |
> > | Energy-FT | 6.34 | 25.06 | 28.37 | 131.99 | 8.44 | 9.30 | 5.14 | 30.66 |
> > | GNNSafe++ | 8.23 | 25.68 | 31.14 | 138.09 | 9.66 | 10.43 | 7.13 | 32.91 |
> > | DeGEM | 11.64 | 15.26 | 15.54 | 149.17 | 10.40 | 17.71 | 5.52 | 32.18 |
> >
> > It can be seen that our method has a similar training cost compared to existing graph-based methods. Notably, our method is even more efficient than GNNSafe++ (the OOD exposure version of GNNSafe) on average.

---

> > > ### Comment · Reviewer_6WFP · 2024-11-27
> > >
> > > Thanks for the detailed response and newly supplemented comparison. I strongly encourage the authors to incorporate the complexity analysis and time/memory cost comparison into the paper which can benefit this work.

---

> > > > ### Author Response · Authors · 2024-11-27
> > > >
> > > > We thank you for acknowledging our work and for raising the score. We will add the additional analysis and comparison in the revision. Thanks again for your time and effort in reviewing our work.

---

### Meta-Review · Area_Chair_RLVL · 2024-12-22

**Metareview:**

To address node-level out-of-distribution (OOD) detection on both homophilic and heterophilic graphs, this paper introduces the Decoupled Graph Energy-based Model (DeGEM). By decoupling the graph encoder from the energy function, DeGEM avoids reliance on homophily assumptions and simplifies MCMC sampling in graph structures. Experiments demonstrate convincing improvements over state-of-the-art methods across diverse datasets, including synthetic, homophilic, and heterophilic graphs.

**Pros:**
- The decoupling of the graph encoder and energy head enables DeGEM to achieve robust performance on heterophilic graphs.
- The model reduces computational overhead of energy-based models by moving MCMC sampling to the latent space.
- DeGEM effectively detects OOD nodes without requiring exposure to OOD data during training.
- Source code is provided to ensuring reproducibility.

**Cons:**
- The model features a relatively complex structure, with multiple components working together, increasing implementation and interpretation difficulty.
- Although ablation studies evaluate various configurations, further analysis of the independent contributions of specific modules (e.g., Conditional Energy) could provide greater clarity.
- While DeGEM excels in heterophilic settings, the theoretical basis for its strong performance on homophilic graphs could be better explained.

DeGEM represents a solid advancement in node-level OOD detection, particularly for heterophilic graphs. Its acceptance is well-supported by all four reviewers.

**Additional Comments On Reviewer Discussion:**

During the rebuttal period, the authors effectively addressed reviewer concerns by providing detailed complexity and scalability analyses, conducting additional experiments on homophily sensitivity, and clarifying DeGEM’s novelty and generality. These responses strengthened reviewer confidence and underscored the model’s robustness and unique contributions to heterophilic graph settings. The authors are expected to incorporate this additional information into the revised paper.

---

### Decision · Program_Chairs · 2025-01-22

Accept (Poster)